



# Hydrological signatures describing the translation of climate seasonality into streamflow seasonality

Sebastian J. Gnann[1], Nicholas J. K. Howden[1], and Ross A. Woods[1]

[1]Department of Civil Engineering, University of Bristol, Bristol, UK

**Correspondence:** Sebastian Gnann (sebastian.gnann@bristol.ac.uk)

**Abstract.** Seasonality is ubiquitous in nature, and it is closely linked to water quality, ecology, hydrological extremes, and water resources management. Hydrological signatures aim at extracting relevant information about hydrological behaviour, and they can be used to better understand hydrological processes and to evaluate hydrological models. Commonly used seasonal hydro-climatological signatures consider climate or streamflow seasonality, but not how climate seasonality translates into
streamflow seasonality. We propose and test hydrological signatures based on the attenuation of the seasonal climate input by a catchment. We approximate the seasonality in the input (precipitation minus potential evapotranspiration) and the output (streamflow) by sine waves. A catchment alters the input sine wave by reducing its amplitude and by shifting its phase. We use these quantities, the amplitude ratio and the phase shift, as seasonal hydrological signatures. We present analytical solutions describing the response of linear reservoirs to periodic forcing to interpret the seasonal signatures in terms of configurations of
linear reservoirs. Using data from the UK and the US, we show that the seasonal signatures exhibit hydrologically interpretable patterns and that they are a function of both climate and catchment attributes. Wet, rather impermeable catchments hardly attenuate the seasonal climate input. Drier catchments, especially if underlain by a productive aquifer, strongly attenuate the input sine wave leading to phase shifts up to several months. Finally, we test whether two commonly used hydrological models (IHACRES, GR4J) can reproduce the observed ranges of seasonal signatures in the UK. The results show that the seasonal
signatures can aid model building and evaluation.

## 1 Introduction

Seasonal patterns are ubiquitous in nature, and many streams have a distinct seasonal flow regime. Streamflow seasonality is primarily driven by climate seasonality and thus sentitive to changes in precipitation, evapotranspiration, and snow fraction (Cayan et al., 1993; Regonda et al., 2005; Berghuijs et al., 2014). The seasonal flow regime is closely linked to water chemistry
and water quality (DeWalle et al., 1997; Vega et al., 1998). Streamflow seasonality plays a crucial role for biological systems and ecosystems (Colwell, 1974; Poff et al., 1997; Poff and Zimmerman, 2010). Low flows are typically seasonal, and droughts – albeit a more general phenomenon than low flows – often occur during the low flow season and thus are to some degree predictable (Smakhtin, 2001; Peters et al., 2003). From a more applied point of view, the seasonal streamflow regime is crucial for water resources management, agriculture, and hydropower generation (Weingartner et al., 2013; Laaha et al., 2013;
Svensson, 2016; Harrigan et al., 2018b). This is reflected in the increased application and development of seasonal forecasting



methods (Shi et al., 2008; Svensson, 2016; Harrigan et al., 2018b). In summary, for many applications the mean seasonal regime is of high importance thus deserves attention.

In this work we focus on the mean seasonal hydrological response. We do not focus, for instance, on the seasonality of events (e.g. storms), noting, however, that the seasonal water balance can have an impact at event scales (Berghuijs et al., 2014). The seasonality of the flow regime is primarily driven by the incoming forcing, that is, the seasonality of precipitation (water) and potential evapotranspiration (energy). Given a certain forcing, the flow regime of a catchment is determined by a catchment's form and function, that is, by how much water can infiltrate, how much water can be stored, and how slowly that water is being released. Since groundwater recharge and thus groundwater discharge are often very seasonal (Jasechko et al., 2014), many hydrogeological studies focus on seasonality, or more specifically on how seasonal recharge is propagated through an aquifer (Townley, 1995; Erskine and Papaioannou, 1997; Peters et al., 2003; Obergfell et al., 2019). Slowly responding groundwater-dominated catchments closely resemble the aquifer system feeding the stream. Understanding the seasonal streamflow regime is therefore crucial for understanding slow (groundwater-driven) dynamics in catchments.

Different aspects of hydrological behaviour, such as streamflow seasonality, can be quantified by summarising metrics now mostly called hydrological signatures (McMillan et al., 2017). The use of such summarising metrics is not new, and they have been used extensively in ecohydrological studies (e.g. Clausen and Biggs, 2000; Olden and Poff, 2003) and hydrological studies (e.g. Jothityangkoon et al., 2001; Farmer et al., 2003). Hydrological signatures offer a way to quantify hydrologic similarity, which makes them useful for catchment classification, for hydrological process exploration, and for predictions in ungauged basins (Sivapalan et al., 2003; Wagener et al., 2007; Hrachowitz et al., 2013; Westerberg et al., 2016). More recently, hydrological signatures have become more popular as a way to guide diagnostic model evaluation (Gupta et al., 2008; Peel and Blöschl, 2011; Euser et al., 2013). Signatures rooted in hydrological theory offer a potentially more meaningful and fit-for-purpose alternative to the typically used statistical metrics such as the Nash-Sutcliffe efficiency (NSE; Nash and Sutcliffe, 1970) or the Kling-Gupta efficiency (KGE; Gupta et al., 2009).

There are many hydrological signatures and we therefore need guidelines for signature selection (McMillan et al., 2017; Addor et al., 2018). Some of these guidelines refer to more technical aspects: the uncertainty in a signature should not be larger within a catchment than between catchments (identifiability), a signature should be insensitive to the data sources (robustness), and a signature should be comparable across (heterogeneous) catchments (consistency; McMillan et al., 2017). When using combinations of signatures, the different signatures should also contain different information, i.e. they should not be redundant (Olden and Poff, 2003; Addor et al., 2018). From a more hydrological perspective, a signature should be meaningful at the relevant scale (representativeness) and a signature should relate to and increase our knowledge of hydrological function (discriminatory power; McMillan et al., 2017). The latter aspect was, for example, highlighted by Addor et al. (2018), who stated that "signatures directly related to the water balance are already well explained by climatic indices" and that other "poorly-predicted signatures deserve more attention". Besides modelling (hydro-)climatic signatures such as the mean flow correctly, we should try to explain and use signatures that tell us more about catchment functioning.

There is a multitude of signatures focusing on seasonality. Climate seasonality is accounted for by (hydro-)climatic signatures such as the (co-)seasonality of precipitation and potential evapotranspiration (Milly, 1994; Knoben et al., 2018). Stream-





flow seasonality can be characterised by the Pardé coefficients (Weingartner et al., 2013) or the regime curve, which is related to the slow flow component of the flow duration curve (FDC; Yokoo and Sivapalan, 2011). Seasonal signatures related to streamflow timing are the half flow date and the half flow interval (Court, 1962), and the date of each annual one-day maximum (or mininum; Richter et al., 1996). Seasonal streamflow signatures focusing on low flows are for example the seasonality

index, which measures the mean day of low flow occurrence and the intensity of seasonality, or the seasonality histogram, which shows the occurence of low flows in each month (Laaha and Blöschl, 2006). Colwell's predictability is another measure describing periodic signals (Colwell, 1974), mostly used in ecological studies. It consists of constancy (how variable the intra-annual flow regime is) and contingency (how persistent the inter-annual flow regime is). All of these signatures describe (parts of) the seasonality of either climate or streamflow, yet none of them look at how climate seasonality translates into streamflow

seasonality. As the transformation of climate input into streamflow is, after all, what we are trying to understand, investigating the seasonal aspect of that seems worthwhile. Relating streamflow to climate input also removes the arbitrariness of picking a start date (e.g. by defining a water year), which is a limitation of many signatures that relate flows to a date (e.g. the half flow date). Furthermore, a signature describing how climate seasonality is translated into streamflow seasonaltiy adds a timing component with a focus on seasonal and thus slower dynamics. This might make it a valuable addition to other (slow flow)

signatures such as the baseflow index (BFI), or the flow duration curve and parts thereof (e.g. $Q_{95}$), which focus on volumes and frequencies, respectively.

In this work, we propose and test the use of hydrological signatures based on how catchments attenuate the seasonal climate input (forcing). We approximate the input signal to a catchment (the forcing) by precipitation minus potential evapotranspiration and the output signal from a catchment by streamflow. We quantify the seasonality in both signals by fitting sine waves to

them. As the period is fixed (one year), the incoming sine wave and the outgoing sine wave differ only in their amplitude, their phase and their mean. As the mean is rather a measure of the annual water balance, we are primarily interested in amplitude and phase. The differences in amplitude and phase are used as signatures describing the steady-state response of a catchment to periodic forcing. This idea is not new. It is similar to the approach of Peters et al. (2003) who investigated drought propagation through groundwater using sinusoidal recharge, and to the approach of Obergfell et al. (2019) who used the seasonal behaviour

as an additional signature in recharge estimation. The approach is also similar to approaches in transit time modelling (e.g. McGuire and McDonnell, 2006; Kirchner, 2016). Instead of focusing on the velocity of water particles, we, however, focus on the hydraulic response to periodic forcing, that is the celerity of the input "wave" of hydraulic potential (Harman, 2019). The proposed signatures are essentially also spectral domain signatures (Montanari and Toth, 2007), focusing only on a certain meaningful period – the annual period.

The seasonal signatures can be related to conceptual linear reservoirs (this will be outlined in Section 2), i.e. they can be interpreted in terms of simple conceptual model structures and parameter values (the reservoir time constants or response times). This gives them some hydrological interpretability (cf. discriminatory power; McMillan et al., 2017). These signatures do not require any parameters, they can be estimated directly from precipitation, potential evapotranspiration and streamflow data. The signatures can be related to the predictability measures proposed by Colwell (1974). The amplitude might be compared to





the constancy (how variable the signal is), and the goodness of fit of the sine wave to the contingency of a signal (how much of the variability can be explained by the seasonal component alone).

In the following, we will first define the seasonal signatures, and we will present analytical solutions describing the response of linear reservoirs to periodic forcing (Section 2). Second, we will calculate the seasonal signatures for a range of catchments in the UK and in the US (Section 4, the data sources are presented in Section 3). We will explore how they relate to hydro-

climatic forcing and catchment form, and we will interpret the underlying hydrological processes (Section 5). Third, we will test whether the seasonal signatures are useful in modelling practice, i.e. we will investigate whether two commonly used hydrological models (IHACRES, GR4J) can reproduce the seasonal signatures (Section 5.4).

## 2 Methods

### 2.1 Extracting seasonal components from time series

#### 2.1.1 Quantification of periodic components

To analyse the periodic components of time series we first need to quantify these components. While we could investigate the whole frequency spectrum of our time series and see how this is altered by a catchment (Montanari and Toth, 2007), we will focus on a period $T$ of one year. The annual period has a clear physical meaning as it is the period the Earth moves in its orbit around the Sun, which is directly linked to the energy input to the Earth system. The input to a catchment, the

forcing $F$, is approximated by precipitation $P$ minus potential evapotranspiration $E_p$ ($F = P - E_p$). We use $E_p$ to avoid the need for a model or additional data which would be needed to obtain actual evapotranspiration $E_a$. This might be particularly problematic in water-limited catchments, where actual evapotranspiration is much smaller than potential evapotranspiration, and in catchments where precipitation and potential evapotranspiration are out of phase. We will discuss that in Section 5. The seasonal component of the forcing $F_{\text{sin}}$ is given by (Milly, 1994):

$$F_{\text{sin}} = \bar{F}\left(1 + \delta_F \sin\left(\frac{2\pi}{T}t + \phi_F\right)\right) \tag{1}$$

where $\bar{F}$ is the mean, $\delta_F$ is the ratio between the amplitude and the mean (the dimensionless amplitude), and $\phi_F$ is the phase (with respect to a reference date) of the seasonal forcing component. The output from a catchment is approximated by streamflow $Q$. The seasonal component of streamflow $Q_{\text{sin}}$ is given by:

$$Q_{\text{sin}} = \bar{Q}\left(1 + \delta_Q \sin\left(\frac{2\pi}{T}t + \phi_Q\right)\right) \tag{2}$$

where $\bar{Q}$ is the mean, $\delta_Q$ is the ratio between the amplitude and the mean, and $\phi_Q$ is the phase (with respect to the same reference date) of the seasonal streamflow component.

Since we know the period $T$ of interest, we need to quantify the mean, the amplitude and the phase of the periodic components. There are different methods to fit a sine curve of a certain period to data. We have compared two sine curve fitting methods leading to virtually the same results (see Supplement for more information on sine curve fitting). For the rest of the

analysis, we will use results obtained by means of multiple linear regression.





### 2.1.2 Calculation of seasonal signatures

Once we have extracted the seasonal components from our time series (precipitation minus potential evapotranspiration, streamflow), we can quantify how the outgoing sine wave $Q_{\sin}$ has been altered by the catchment by comparing it to the incoming sine wave $F_{\sin}$. We define two metrics, the amplitude ratio and the phase shift, which together we call seasonal sig-

natures. The amplitude ratio $A$ is the ratio between the seasonal streamflow amplitude $\delta_Q \bar{Q}$ and the seasonal forcing amplitude $\delta_F \bar{F}$:

$$A = \frac{\delta_Q \bar{Q}}{\delta_F \bar{F}} \tag{3}$$

Given a closed long-term water balance, the amplitude ratio should theoretically always be between zero and unity, that is, the streamflow amplitude cannot be larger than the forcing amplitude. The phase shift $\phi$ is the difference between the phase of the

seasonal streamflow component $\phi_Q$ and the phase of the seasonal forcing component $\phi_F$:

$$\phi = \phi_Q - \phi_F \tag{4}$$

The phase shift should theoretically always be positive (the input should lead the output) and smaller than one year.

## 2.2 Linear reservoir theory

A linear reservoir is described by:

$$Q = \frac{S}{\tau} \tag{5}$$

where $Q$ [mm d$^{-1}$] is the outflow from the reservoir, $S$ is storage [mm] and $\tau$ [d] is a time constant describing how fast (slow) the reservoir responds. Conservation of mass requires:

$$\frac{dS}{dt} = Q_{\text{in}} - Q \tag{6}$$

where $Q_{\text{in}}$ is the inflow to the reservoir.

### 2.2.1 Periodic forcing of a linear reservoir

If we approximate the seasonal input to a linear reservoir by a sine wave of period $T$ (e.g. one year), we can combine Equations (1), (5) and (6) to obtain:

$$\frac{dQ_{\sin}}{dt} = \frac{\bar{F}}{\tau}\left(1 + \delta_F \sin\left(\frac{2\pi}{T}t + \phi_F\right)\right) - \frac{Q_{\sin}}{\tau} \tag{7}$$

We might neglect the (initial) phase if we choose a starting time $t$ that is aligned with the seasonal forcing component ($\phi_F = 0$).

It can be shown that the steady state response of a linear reservoir to a sinusoidal input signal is a damped and phase shifted version of the input signal (see Supplement for a more detailed derivation; or Eriksson, 1971; Peters et al., 2003):

$$Q_{\sin}(t) = \bar{F}\left(1 + \delta_F A \sin\left(\frac{2\pi}{T}t + \phi\right)\right) \tag{8}$$





where $A$ is the amplitude ratio and $\phi$ is the phase shift induced by a single linear reservoir.

$$A = \frac{1}{\sqrt{1 + (2\pi\frac{\tau}{T})^2}} \tag{9}$$


$$\phi = \arccos\left(\frac{1}{\sqrt{1 + (2\pi\frac{\tau}{T})^2}}\right) = \arccos(A) \tag{10}$$

We can rewrite Equation (8) as follows:

$$Q(t) = \bar{Q}\left(1 + \delta_Q \sin\left(\frac{2\pi}{T}t + \phi\right)\right) \tag{11}$$

In a mass conserving system in steady-state, the mean of the output should equal the mean of the input. If the means obtained
from data are different, either the forcing term is inaccurate (e.g. due to differences between actual and potential evapotranspi-
ration) or the streamflow term is inaccurate (e.g. due to other losses or gains). The product of input amplitude and amplitude
ratio equals the output amplitude ($\delta_F \bar{F} A = \delta_Q \bar{Q}$).

From Equations (9) and (10), we can see that the amplitude ratio and the phase shift are given by $A$ and $\arccos(A)$, respec-
tively. Since $A$ is fully defined by the ratio between $\tau$ and $T$, and $T$ is usually known (e.g. one year), we can theoretically use $A$
to determine the time constant $\tau$ of the reservoir. This requires the identification of both the seasonal components of the input
and output signal of that period (see Section 2.1), and assumes the system to behave as a single linear reservoir.

The amplitude ratio $A$ and the phase shift $\arccos(A)$ can be plotted against each other for various values of $\tau$ as shown
in Figure 1. This results in a characteristic curve which captures the response of all single linear reservoirs. Different time
constants $\tau$ (as proportions of the period, here one year) lead to different positions on the curve. For very fast reservoirs, the
phase shift is close to 0 days and the amplitude ratio is close to unity (that is, the signal is not attenuated at all). For very slow
reservoirs, the signal is phase shifted up to 91 days and the amplitude ratio approaches 0. The maximum phase shift of about 91
days corresponds to a quarter of a period (90 degrees). Mathematically, this can be explained by Equation (10), as the arccosine
of a quantity between 0 and unity (such as $A$) ranges between 0 and 90 degrees.

Note the similarity of Figure 1 to Figure 3c in Kirchner (2016), which shows the relationship between phase shift and
amplitude ratio for gamma-distributed catchment transit time distributions. An exponential transit time distribution (a special
case of the gamma distribution) corresponds to a linear reservoir describing the velocity of particles. Similarly, a linear reservoir
describing the impulse response (the linear reservoir from Equation (5)), i.e. the celerity of the incoming wave of hydraulic
potential, corresponds to an exponential response time distribution or an exponential unit hydrograph (cf. Eriksson, 1971;
Dooge, 1973).

### 2.2.2 Combinations of linear reservoirs

Linear systems (Dooge, 1973) have the advantage that it is relatively straightforward to add more components, that is, reser-
voirs. It is quite common to have serial and/or parallel combinations in rainfall-runoff models. In theory, we can find analytical



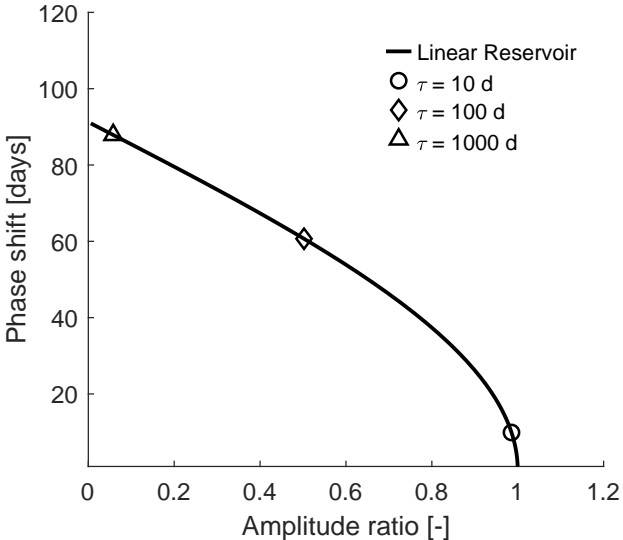

**Figure 1.** Amplitude ratio against phase shift for a single linear reservoir for varying time constants $\tau$. Three example time constants are indicated by the symbols.

solutions for the amplitude ratio and phase shift for all combinations of linear reservoirs (cf. to the transfer function approach of Young, 1998, who identify combinations of reservoirs that fit the data best in an inductive way). There are two basic arrangements, a serial arrangement of reservoirs and a parallel arrangement of reservoirs.

### 2.2.3 Linear reservoirs in series

Linear reservoirs in series can be conceptualised as follows. Every outflow is the inflow to the next reservoir. Hence, if the $i$-th reservoir has a time constant $\tau_i$, the amplitude ratios $A_i$ are multiplied and the phase shifts $\phi_i$ are added (see Supplement for a more detailed derivation):

$$A_{\text{tot}} = \prod_{i=1}^{n} A_i \qquad (12)$$

$$\phi_{\text{tot}} = \sum_{i=1}^{n} \phi_i = \sum_{i=1}^{n} \arccos(A_i) \qquad (13)$$

Figure 2 shows the amplitude ratio plotted against the phase shift similar to Figure 1, but now with two linear reservoirs in series. The different lines are examples with fixed time constants of the first reservoir. They all start from the black line (from the points marked by the symbols in Figure 1), the characteristic curve for a single linear reservoir, which is the lower limit. Then, as the time constant of the second reservoirs increases, the lines "move" left and upwards, which corresponds to a decrease in amplitude ratio and an increase in phase shift. For example, the red line ($\tau_1 = 10$ d) starts out with a phase shift of about 10 days, and ends at a phase shift of about 101 days, which is an increase of about 91 days, the maximum phase



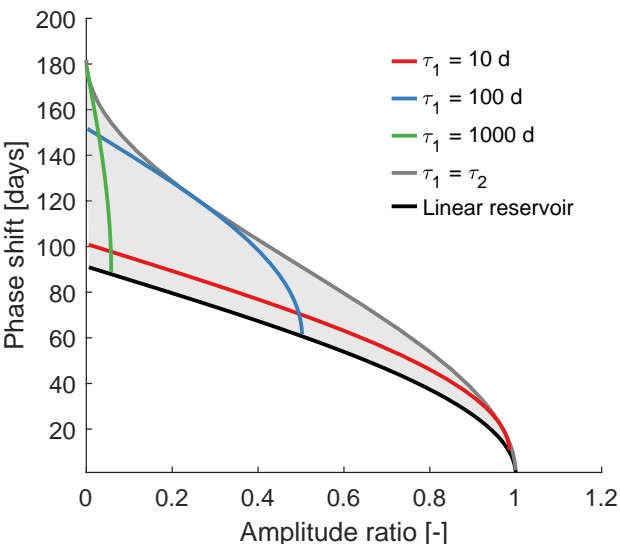

**Figure 2.** Amplitude ratio against phase shift for two linear reservoirs in series. Each line corresponds to a fixed time constant for the first reservoir ($\tau_1$), while the time constant of the second reservoir varies (1 d $\leq \tau_2 \leq$ 10000 d; it is increasing from right to left). The black line indicates a single linear reservoir (the lower boundary). The grey line indicates the upper boundary where $\tau_1 = \tau_2$. The shaded area contains all possible combinations of amplitude ratio and phase shift for two linear reservoirs in series.

shift of the second reservoir. The lines cross each other as we allow $\tau_2$ to be larger than $\tau_1$. This implies that sometimes a faster reservoir is followed by a slower one, and sometimes a slower reservoir is followed by a faster one. The grey shaded area contains all possible combinations for two reservoirs in series. The lower limit is a single linear reservoir. The upper limit corresponds to two reservoirs with the same time constant (a two-reservoir Nash cascade), which equals a gamma distribution with a shape parameter equal to 2.

### 2.2.4 Linear reservoirs in parallel

Linear reservoirs in parallel result in a "mixture" of the outflows from each reservoir. The resulting flow is a combination of sine waves of the same period, weighted by the fraction $p_i$ going into each reservoir. For the sake of simplicity, we only consider two reservoirs in parallel. We denote the fraction going into the second reservoir by $p$, and therefore the fraction going into the first reservoir by $1-p$. Thinking of the second reservoir as the slow one, $p$ might be compared to the idea of the baseflow index (BFI). For two reservoirs in parallel we get (see Supplement for a more detailed derivation):

$$A_{\text{tot}} = \sqrt{\left[(1-p)A_1\cos\phi_1 + pA_2\cos\phi_2\right]^2 + \left[(1-p)A_1\sin\phi_1 + pA_2\sin\phi_2\right]^2} \tag{14}$$

$$\phi_{\text{tot}} = \arctan\left(\frac{(1-p)A_1\sin\phi_1 + pA_2\sin\phi_2}{(1-p)A_1\cos\phi_1 + pA_2\cos\phi_2}\right) \tag{15}$$





**Figure 3.** Amplitude ratio against phase shift for two linear reservoirs in parallel. **(a)** Each line has a fixed time constant for the first reservoir ($\tau_1$), while the time constant of the second reservoir varies ($10\,\mathrm{d} \leq \tau_2 \leq 10000\,\mathrm{d}$; it is increasing from right to left). The fraction $p$ going into the second reservoir is 0.3. **(b)** Same as **(a)** with $p = 0.6$. **(c)** Same as **(a)** with $p = 0.9$. **(d)** Each line has a fixed time constant for the first reservoir ($\tau_1 = 1\,\mathrm{d}$), and for the second reservoir ($\tau_2$). The fraction $p$ going into the second reservoir is varied (it is increasing from right to left). The shaded area contains all the possible combinations of amplitude ratio and phase shift for two linear reservoirs in parallel.





Figure 3 shows the amplitude ratio plotted against the phase shift similar to Figure 1, but now with two linear reservoirs in parallel. We show multiple plots to highlight the three degrees of freedom: the two reservoir time constants and the fraction going into each reservoir. The latter is highlighted in Figure 3d, but also visible in Figures 3a-c. We can see that there is an "extreme case" where the output signal effectively comes from the first reservoir only as the second reservoir is so slow that

it hardly contributes to the resulting sine wave. As only $1 - p$ of the total input goes into the first reservoir, the amplitude will be about $1 - p$ times the input amplitude. The grey shaded area contains all the possible combinations for two reservoirs in parallel. The upper limit is a single linear reservoir. The lower limit is effectively given by the $x$- and the $y$-axis.

## 2.3 Seasonal signatures as a diagnostic tool for evaluating hydrological models

We use two conceptual rainfall-runoff models and we use the seasonal signatures as a diagnostic tool to assess their performance

(Gupta et al., 2008). In particular, we test whether the models are generally capable of reproducing the range of observed seasonal signatures (cf. Vogel and Sankarasubramanian, 2003). We limit the analysis to two models and 40 catchments to keep the computational demand manageable. The catchments are described in Section 3.

The first model is the IHACRES model. It is conceptually relatively similar to the considerations in Section 2. It has a soil moisture store (non-linear deficit store), and two parallel linear stores for fast flow and for slow flow (Croke and Jakeman,

2004). It has been used in many modelling studies in Australia (Post and Jakeman, 1999) and also in the UK (Sefton and Howarth, 1998). The second model is the GR4J model. It also has a parallel flow structure, but the internal parametrisation is different. It contains more non-linearities and it has fixed internal parameters. Additionally, it has a groundwater exchange parameter aimed at representing inter-catchment groundwater flows. It has been used in many modelling studies in France (Perrin et al., 2003), in the UK (Smith et al., 2019; Harrigan et al., 2018b) and in the US (Oudin et al., 2018). We use the

implementations of the two models in the MARRMoT toolbox v1.2 (Knoben et al., 2019a), a Matlab toolbox containing many hydrological models aimed at model comparison studies. The pure delay function in the MARRMoT implementation of IHACRES is set to 0, making it (conceptually, not necessarily numerically) equal to the version used by Croke and Jakeman (2004). In our model evaluation, IHACRES has therefore 6 parameters, and GR4J has 4 parameters.

To test which ranges of seasonal signatures the two models can reproduce, we run a Monte-Carlo sampling experiment. We

sample parameter sets using Latin Hypercube sampling and we test an increasing number of parameter sets to see whether the results are robust (see Supplement). We mostly use the recommended parameter ranges from the MARRMoT toolbox, which are intended to be wide ranges. We use a narrower range for the fast flow routing delay (1 to 5 days), to ensure that it is indeed "fast flow". We then use the modelled streamflow time series to calculate three hydrological signatures: the two seasonal signatures presented here and the baseflow index (BFI). These are explored in a rather general way, as we want to

examine what the models can do without actually calibrating them to certain catchments (cf. Vogel and Sankarasubramanian, 2003).

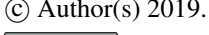



**Table 1.** Hydrological signatures and catchment attributes used in this study.

| Name | Description | Unit | Range | Reference |
|---|---|---|---|---|
| Hydrological signatures | | | | |
| BFI | Baseflow index | [-] | $[0,1]$ | Institute of Hydrology (1980) |
| $A$ | Amplitude ratio | [-] | $[0,1]$[1] | Equation (3) |
| $\phi$ | Phase shift | [d] | $[0,365]$[2] | Equation (4) |
| Catchment attributes | | | | |
| $E_p/P$ | Aridity index | [-] | $[0,\infty]$ | |
| $I_m$ | Moisture index | [-] | $[-1,1]$ | Knoben et al. (2018) |
| $I_{m,r}$ | Moisture index seasonality | [-] | $[0,2]$ | Knoben et al. (2018) |
| $f_s$ | Snow fraction | [-] | $[0,1]$ | Knoben et al. (2018) |
| PROPWET | Catchment wetness index | [-] | $[0,1]$ | National River Flow Archive (2019) |
| % fractured aquifer | Fraction of highly productive fractured aquifer | [%] | $[0,100]$ | National River Flow Archive (2019) |
| % carbonate rock | Fraction of carbonate sedimentary rock | [%] | $[0,100]$ | Addor et al. (2017) |

[1] Should in theory be smaller than unity. [2] Should theoretically always be positive and in practice be smaller than one year. Further discussions on the possible ranges of the seasonal signatures can be found in the text.

## 3 Data

### 3.1 Data sources

We use catchment data from Great Britain and the United States. The data for the UK is obtained from different sources.
Daily streamflow data, catchment characteristics and catchment boundaries are obtained from the NRFA (National River Flow Archive, 2019), precipitation data from CEH-GEAR (Tanguy et al., 2016), and potential evapotranspiration data from CHESS-PE (Robinson et al., 2016). For the model evaluation we select catchments that are part of the UK Benchmark Network (Harrigan et al., 2018a), which describes catchments in the UK that are near-natural. The subset of catchments is chosen to be representative of the UK, details are shown in the Supplement. The data for the US is obtained from the CAMELS dataset
(Newman et al., 2015; Addor et al., 2017). CAMELS includes daily precipitation, potential evapotranspiration and streamflow data as well as a wide range of catchment attributes for 671 catchments in the contiguous US. We trim the daily data to contain only full water years (starting 1 October) and we analyse data from 1989 to 2009. While we need to pick a start date for the analysis, this date does not influence the results (e.g. using 1 January as starting date would result in the same phase shift).

### 3.2 Hydrological signatures and catchment attributes

We calculate different hydrological signatures and we use different catchment attributes, all summarised in Table 1.





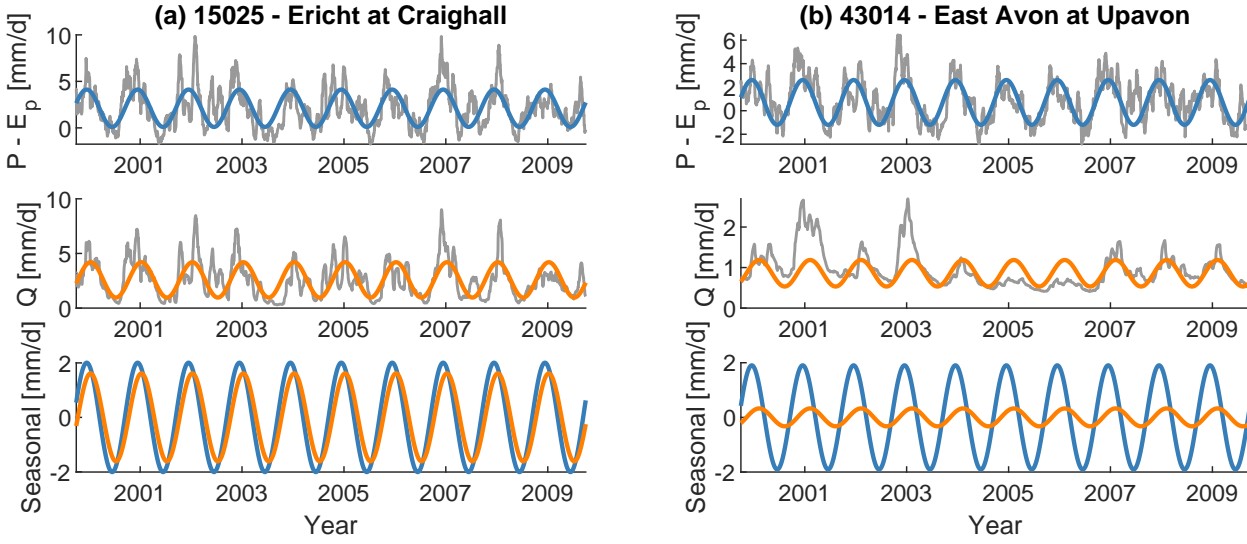

**Figure 4.** Climate input ($P$ - $E_p$) and catchment output ($Q$) for two catchments in the UK, and their respective seasonal components. The time series are smoothed using a 30-day moving mean. The Ericht is a rather responsive catchment (BFI = 0.47), while the East Avon has a large baseflow component (BFI = 0.89). Note that for the bottom plots ("Seasonal") the mean values of the sine curves are set to zero.

## 4  Results

### 4.1  Extracting seasonal components from time series

First, we extract seasonal components from $P - E_p$ (forcing) and $Q$ (streamflow) for all catchments. The resulting sine wave parameters are then used to calculate the amplitude ratios (Equation (3)) and phase shifts (Equation (4)), respectively. Figure
4 shows $P - E_p$ and $Q$ for two catchments alongside their seasonal (sinusoidal) components. Both catchments experience a similar forcing, but their response is very different. The Ericht at Craighall, a rather responsive catchment, shows a seasonal streamflow component that is very similar to the seasonal forcing component. In contrast, the East Avon at Upavon, a groundwater-dominated catchment, shows a strongly attenuated seasonal streamflow component. For our seasonal signatures this would mean (a) that the responsive catchment has a high amplitude ratio, i.e. the streamflow amplitude is almost as large
as the forcing amplitude, while the groundwater-dominated catchment has a low amplitude ratio. And (b) that the responsive catchment has a small phase shift, i.e. it responds quickly to the (seasonal) forcing, while the groundwater-dominated catchment has a large phase shift.





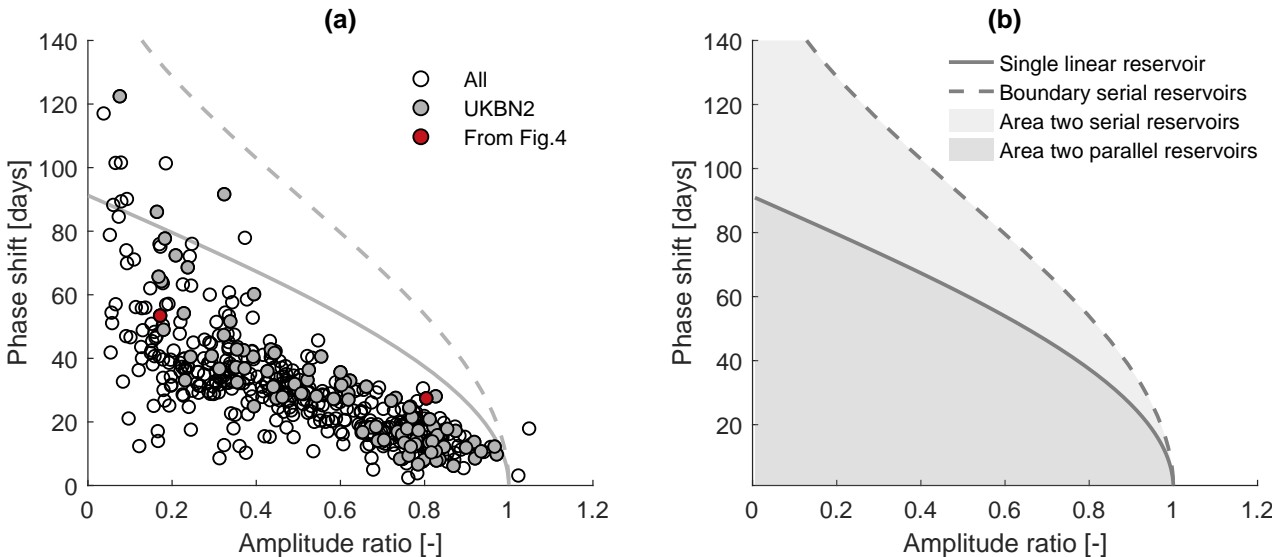

**Figure 5. (a)** Amplitude ratio against phase shift for UK catchments. Grey dots indicate benchmark catchments, red dots indicate the two catchments shown in Figure 4. Grey solid line indicates a single linear reservoir, grey dashed line indicates the outer envelope for two reservoirs in parallel. Note that both axes are limited (two catchments are not shown). **(b)** Theoretical areas and limits for single linear reservoir, two reservoirs in series, and two reservoirs in parallel.

## 4.2 Seasonal signatures of observed catchment data

To visualise the seasonal signatures, we plot the amplitude ratios and phase shifts in a similar way as in Figures 1, 2, and
3. This is shown in Figure 5a for all UK catchments. These include catchments with human influences, such as groundwater abstractions, man-made reservoirs or water transfers. The overall pattern in Figure 5a is very similar to the pattern using benchmark catchments alone (grey dots). We therefore use all of the catchments, noting that a few catchments might be unsuitable for individual analyses.

Figure 5a shows that most of the catchments fall below the solid grey line, the line which indicates the type of response
that could be simulated by a single linear reservoir (see Figure 5b). The area below the solid line can be simulated by two reservoirs in parallel. This would be the most parsimonious way to reproduce the observed behaviour if we decide to construct our model using linear reservoirs only. A few catchments plot above the solid line. For these catchments, the most parsimonious way to repdroduce the pair of observed amplitude ratio and phase shift would therefore be two reservoirs in series. Very few catchments have an amplitude ratio larger than unity. While this could be caused by various errors in the data, it is likely due to
erroneous catchment areas and/or the presence of inter-catchment groundwater flows or water transfers. If a catchment receives more rainfall than the surface catchment area suggests (runoff ratio > 1), the amplitude in the output signal (streamflow) can be larger than the amplitude in the (erroneous) input signal.





### 4.2.1 Relationship between seasonal signatures and catchment attributes – UK

Figure 6 shows pairs of amplitude ratios and phase shifts, coloured according to different hydrological signatures and catchment attributes, respectively. Figure 6a shows a clear pattern between the moisture index and the seasonal signatures. Generally, the less humid the catchments (indicated by a more intense red colour), the lower the amplitude ratio and the larger the phase shift. In other words, drier catchments attenuate the incoming forcing signal more strongly. This might partly be because we use potential evapotranspiration as our forcing. Lower actual evapotranspiration than potential evapotranspiration leads to a decreased input amplitude and thus to a higher amplitude ratio. Very humid catchments plot close together and the relationship

between amplitude ratio and phase shift seems to be almost linear. Less humid catchments (note that in the UK none of the catchments are actually water-limited on an annual scale) show a larger spread, especially regarding the phase shift. Figure 6b shows a very similar pattern between the catchment wetness index and the seasonal signatures. Wetter catchments exhibit higher amplitude ratios and lower phase shifts, and vice versa. The catchment wetness index is strongly correlated with the moisture index (Spearman rank correlation of 0.94). Figure 6c shows a clear pattern between the baseflow index and the

seasonal signatures. In contrast to the moisture index, where the stratification follows mostly the $x$-axis (amplitude ratio), the stratification follows mostly the $y$-axis (phase shift). Catchments with high BFIs exhibit low amplitude ratios and large phase shifts, and vice versa. Finally, in Figure 6d we can see that catchments underlain by highly productive fractured aquifers exhibit (with a few exceptions) low amplitude ratios and large phase shifts.

### 4.2.2 Relationship between seasonal signatures and catchment attributes – US

Figure 7 shows pairs of amplitude ratios and phase shifts for the US, coloured according to different hydrological signatures and catchment attributes, respectively. Catchments with significant snow fraction ($f_s > 0.001$) are removed, as snow presents another hydrological process which is not the focus of this study. Generally, snow adds another storage process, and this is reflected in large phase shifts observed in snowy catchments (see Supplement for more information). The non-snowy catchments in the US show a similar trend to the catchments in the UK. Yet generally, the amplitude ratios are lower and the phase shifts

larger compared to the UK (note that the $y$-axes in Figure 7 differ in their range from the $y$-axes in Figure 6). Humid catchments tend to have higher amplitude ratios and smaller phase shifts (Figure 7a). Climate seasonality, indicated by the moisture index seasonality (see Figure 7b), also influences the seasonal signatures. Catchments with a larger moisture index seasonality, i.e. a more variable monthly moisture index over the year, tend to have smaller phase shifts. The BFI (Figure 7c) does not show such a clear pattern as for the UK catchments (Figure 6c). Similarly, subsurface properties such as the fraction of carbonate

sedimentary rock (Figure 7d; and other attributes not shown here) only show a weak relationship with the seasonal signatures. Catchments with larger fractions of carbonate sedimentary rocks tend to have lower amplitude ratios and larger phase shifts. The overall pattern, however, is rather scattered. Contrary to the UK, some of the catchments in the US plot outside the area that can be modelled by either two reservoirs in series or in parallel and some catchments have phase shifts larger than 182 days, the approximate limit for two reservoirs in series. These catchments are very arid and the low moisture seasonality index

indicates that most the precipitation in these catchments falls when potential evapotranspiration is highest, i.e. in summer.





**Figure 6.** Amplitude ratio against phase shift for UK catchments. Grey solid line indicates a single linear reservoir, grey dashed line indicates the outer envelope for two reservoirs in parallel. Colours indicate **(a)** the moisture index, **(b)** the catchment wetness index, **(c)** the baseflow index, and **(d)** the fraction of highly productive fractured aquifer. Note that both axes are limited (two catchments are not shown).

### 4.3 Seasonal signatures as a diagnostic tool for evaluating hydrological models

In a similar fashion as for the observed catchment data, we now investigate the model runs using IHACRES and GR4J. Figure 8 shows the resulting amplitude ratios and phase shifts for all model runs, that is for 20000 parameter sets using data from a subset of 40 catchments. These plots show which combinations of the seasonal signatures (and BFI) can be obtained with each model, given the forcing of 40 different catchments covering most of the hydro-climatic variability of the UK. They hence show the "signature space" of a model in the dimensions given by amplitude ratio and phase shift (and BFI). IHACRES







**Figure 7.** Amplitude ratio against phase shift for CAMELS catchments. Catchments with snow fraction ($f_s > 0.001$) are removed from the analysis. Grey solid line indicates a single linear reservoir, grey dashed line indicates the outer envelope for two reservoirs in parallel. Colours indicate **(a)** the moisture index, **(b)** the moisture index seasonality, **(c)** the baseflow index, and **(d)** the fraction of carbonate sedimentary rock. Note that both axes are limited (13 catchments are not shown) and that the range of the phase shift-axis is different from Figure 6.

(Figure 8a) shows a pattern that covers the area that can be reproduced by two reservoirs in parallel, and most of the area that can be reproduced by two reservoirs in series (see Figures 2 and 3). IHACRES sometimes yields phase shifts that are close to one year (not shown here), which are effectively negative phase shifts. Negative implies that the periodic component of $Q$

leads the periodic component of $P - E_p$. This can happen if actual evapotranspiration $E_a$ differs considerably from potential evapotranspiration $E_p$, and hence most of the input seasonality stems from $P$ (and not $E_p$). This can be observed in a few catchments in the US (not shown here). It is only observed once in the UK (in a catchment with a man-made reservoir, not shown here), and therefore we do not investigate these model runs further. GR4J (Figure 8b) covers a rather different signature





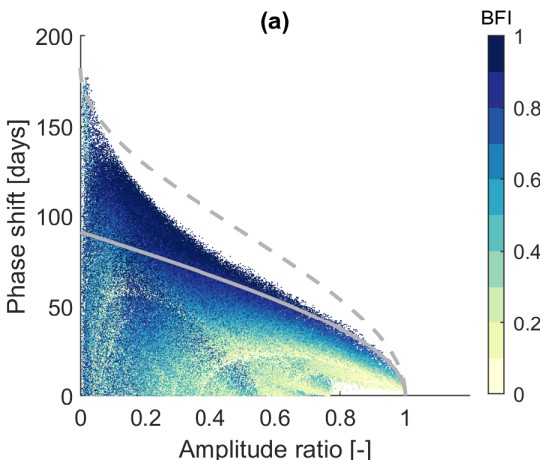
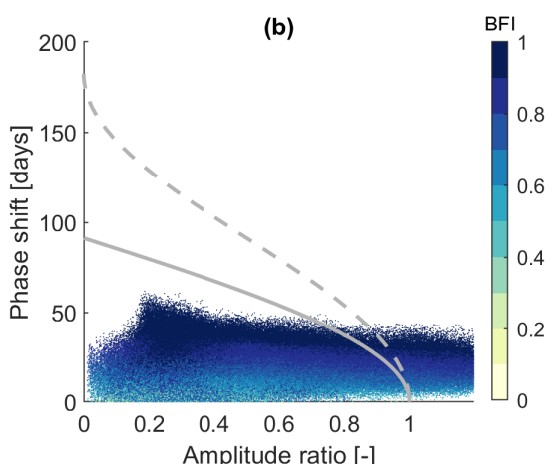

**Figure 8.** Amplitude ratio against phase shift for 40 catchments using 20000 parameter sets each for **(a)** IHACRES and **(b)** GR4J. Note that both axes are limited.

space. The phase shift never exceeds 60 days and the amplitude ratio often exceeds unity. Amplitude ratios larger than unity

are due to the groundwater exchange parameter, which allows the model to import water in addition to incoming $P$.

Figure 9 shows one-dimensional signatures spaces for three hydrological signatures from the two hydrological models. The plots are created as follows. For every catchment and for each of the signatures, we summarise the resulting signature values by a probability density function (PDF). For example, running IHACRES with 20000 parameter sets for one catchment leads to 20000 values for the BFI. We use these 20000 values to fit a PDF via kernel density estimation, resulting in one curve per

signature per catchment (a proxy for the histogram). Plotting curves for all catchments on top of each other shows the range and likelihood of a signature value being produced by a certain model structure (using a certain set of parameter sets). How much the curves differ from each other gives an indication of how a change in forcing changes the resulting signature distribution. The forcing is indicated by the colour of a curve, which corresponds to a certain aridity index. These plots thus tell us which signature values a model structure tends to produce (given a certain sampling scheme) and how (much) a signature varies

with varying forcing. Note that this evaluation is independent of observed streamflow, it just shows the theoretically possible hydrological response that a model can simulate. By comparing observed hydrological signatures to this theoretical signature space, we might be able to tell whether a certain model structure is generally able to simulate a certain set of catchments.

For the BFI (Figures 9a,b) we can see that IHACRES covers the whole possible space (0 to 1) relatively evenly, and it does not vary substantially with varying forcing. This means that the modelled BFI is essentially a function of the model (parameters)

alone. GR4J tends to produce very high BFIs for almost every parameter set. BFI values below 0.5 are possible with GR4J, but rather rare (or unlikely). The forcing has a secondary influence on the BFI, with more humid catchments leading to lower BFIs. Figures 9c-f are in line with Figure 8. IHACRES can produce amplitude ratios from 0 to 1 and phase shifts up to about



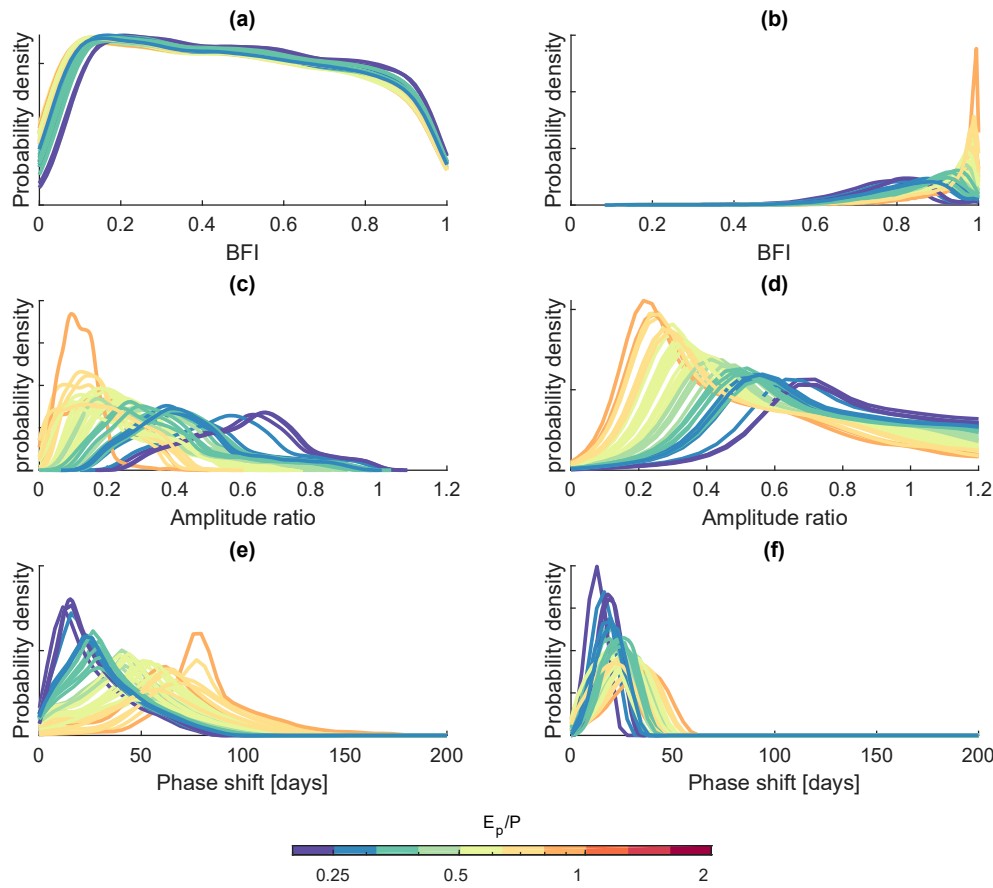

**Figure 9.** PDFs of different hydrological signatures ("signature space") resulting from the parameter sampling experiment for 40 catchments. **(a)** BFI IHACRES, **(b)** BFI GR4J, **(c)** amplitude ratio IHACRES, **(d)** amplitude ratio GR4J, **(e)** phase shift IHACRES, **(f)** phase shift GR4J. The colours indicate the aridity index of the catchments. Note that the $x$-axes are limited and that the $y$-axes are arbitrary and not to scale.

182 days, which is the limit for two reservoirs in series. GR4J can produce amplitude ratios that clearly exceed one, and cannot model phase shifts larger than 60 days – at least given the parameter ranges chosen. For both models, more arid forcing leads to lower amplitude ratios and larger phase shifts, and vice versa.

## 5   Discussion

### 5.1   Representation of seasonal components by sine waves

A sine wave is a simple way of describing the seasonality of a signal. The results suggest that for most of the catchments investigated here, this approach is reasonable and efficient. Figure 4 shows that the average seasonal pattern is captured by the





fitted sine waves. Perhaps more importantly, the patterns visible in Figures 5, 6, and 7 indicate that the extracted seasonal components are meaningful and not just noise. Of course, as with every model, the chosen description of the seasonal components is imperfect. The UK catchments and most of the US catchments exhibit a relatively strong unimodal (climate) seasonality (see e.g. Knoben et al., 2018). In other climates with a less distinct seasonal pattern, or with two seasons per year (Knoben et al., 2019b), our approach will not work. Semi-arid and arid catchments also tend to have a less smooth seasonal input, as water

availability is more fragmented (Peters et al., 2003). They also show a strong difference between potential evapotranspiration and actual evapotranspiration, which might limit the applicability of our approach (we will discuss that later in more detail). We exclude catchments where precipitation is falling as snow. While snowy catchments are typically also strongly seasonal (Schaefli, 2016), this seasonality is mostly a climate phenomenon. It is rather related to temperature seasonality and not to the response of a catchment to periodic forcing.

## 5.2 A perceptual model of the seasonal response of catchments in the UK

The results, in particular Figures 5 and 6, show clear patterns in the seasonal signatures. We can see that the seasonal response in the UK can be simulated by either two reservoirs in series or two reservoirs in parallel. This does not mean that there are no other configurations of more reservoirs leading to the same pairs of amplitude ratio and phase shift. Rather, two reservoirs in series and in parallel, respectively, are the most parsimonious reservoir configuration to reproduce the observed seasonal

behaviour. Of course, two reservoirs in parallel and two reservoirs in series, respectively, might be seen as "special cases" of a soil reservoir followed by a fast and a slow reservoir, i.e. a three-reservoir arrangement. Furthermore, there might be concepts other than reservoirs which are capable of explaining the observed behaviour. Still, the observed patterns, both where the catchments plot in the amplitude-phase shift plot (Figure 5) and how the catchment attributes relate to that (Figure 6), suggest that the seasonal signatures are indeed a window into catchment functioning (Berghuijs et al., 2014) and thus have

discriminatory power (McMillan et al., 2017; Addor et al., 2018).

Figures 6a and 6b show how climate aridity and catchment wetness influence amplitude ratio and phase shift. The observation that more humid catchments respond more quickly to forcing (Figures 6a and 6b) concurs with our understanding of these catchments. Wetter and therefore more saturated catchments partition the incoming water mostly into fast flow. The hydrograph closely resembles the forcing, which can also be seen in Figure 4 for the responsive Ericht river. The drier the catchments

become, the more water is able to infiltrate and subsurface properties become more important. This might explain why the spread becomes larger for less humid and hence less saturated catchments. In less humid catchments, actual evapotranspiration is more likely to deviate from potential evapotranspiration. This might be another reason for the greater attenuation in drier catchments, as the actual input $(P - E_a)$ is lower than the theoretical one we compare to $(P - E_p)$. In the UK the assumption that $E_a = E_p$ seems reasonable (see Supplement for further information). In more arid regions, such as parts of the US (see

Section 5.3), this assumption might be invalid.

The variability among UK catchments that cannot be explained by catchment wetness can mostly be explained by subsurface properties and the associated response time of a catchment. Catchments with high BFIs and thus large baseflow components show lower amplitude ratios and larger phase shifts, that is a more damped and lagged response (Peters et al., 2003). This





can also be seen in Figure 4 for the groundwater-dominated East Avon river. The relationship between BFI and the seasonal
signatures (Figure 6c) is not surprising, yet since the relationship is not unique, the seasonal signatures add another piece of
information. In particular, the phase shift adds a timing component, which quantifies how long – on average – the seasonal
input is delayed to become the seasonal output. While the phase shift is only a few days for the most responsive catchments, in
the slowest catchments the seasonal signal is shifted up to four months. Since the BFI is rather a consequence of a catchment's
hydrological behaviour (as are the seasonal signatures) than an attribute of a catchment, the BFI cannot be seen as a cause
for the observed patterns in the seasonal signatures. It cannot be used, for example, as a predictor in ungauged catchments. A
qualitative attribute that could theoretically be available in ungauged catchments, the fraction of highly productive fractured
aquifer, reinforces the influence of the subsurface (Figure 6d). Except for a few catchments, catchments underlain by such
an aquifer exhibit very large phase shifts. In fact, all the catchments above the single reservoir line are underlain by highly
productive aquifers. In these catchments, mostly underlain by Chalk, almost all the incoming water infiltrates into the aquifer,
and the fast flow component often is negligible. This might explain why they do not behave like reservoirs in parallel, but rather
like reservoirs in series, e.g. a soil reservoir (recharge) and a very slow groundwater reservoir. The few catchments which are
underlain by highly productive aquifers, but do not exhibit large phase shifts, are typically overlain by rather impermeable drift,
which stops water from infiltrating into the aquifer below.

Many models frequently used (and some of them developed) in the UK have a parallel flow structure, and catchments are
usually conceptualised as having a fast and a slow component. While parametrisations and model structures vary between
models, an overall parallel flow structure following a soil moisture module can be found in the PDM model (Moore, 2007), the
TOPMODEL modelling concept (consisting of two fast flow responses; Beven and Kirkby, 1979), the IHACRES model (Croke
and Jakeman, 2004), the GR4J model (Perrin et al., 2003), and many others. These or similar models have been applied to many
catchments in the UK by various authors (e.g. Smith et al., 2019; Lane et al., 2019; Coxon et al., 2019). The seasonal signatures
suggest that for most of the catchments, particularly if they are not underlain by a highly productive aquifer, a parallel model
structure is a reasonable choice (at least for reproducing the response to seasonal forcing). For some groundwater-dominated
catchments, however, the fast flow component seems to be rather unimportant. Many of these catchments, typically catchments
underlain by Chalk, could only be poorly modelled in national-scale modelling studies (Smith et al., 2019; Lane et al., 2019;
Coxon et al., 2019). While this might partly be due to water balance problems (inter-catchment groundwater flows), it might
also be due to an inadequate model structure or inadequate parameter ranges. The most parsimonious reservoir configuration
to explain the seasonal behaviour of these catchments (phase shifts > 91 days) would be two reservoirs in series, e.g. a soil or
unsaturated zone reservoir transforming the incoming forcing into recharge, and a (linear) groundwater reservoir. At least one
of these reservoirs would need to be very slow to obtain such large phase shifts (cf. Figure 2). For these groundwater-dominated
catchments, a serial structure as it is also used in simple lumped groundwater models (e.g. Peters et al., 2003; Obergfell et al.,
2019), seems to be a reasonable choice (at least for reproducing the response to seasonal forcing). As mentioned before, two
reservoirs in parallel and two reservoirs in series, respectively, might be seen as "special cases" of a soil reservoir followed by a
fast and a slow reservoir. For example, some of the catchments underlain by a highly productive aquifer fall in the area that can
be simulated by two reservoirs in parallel (see Figure 6d). Their large phase shifts and their proximity to the "single reservoir



line" suggest, however, that the slow flow component is of particular importance and that large time constants (> 100 days) are
required to model their behaviour.

    In summary, the first control on the attenuation of the seasonal signal in the UK is the partitioning between fast flow and
slow flow. More saturated catchments partition more rainfall into fast flow and hence lead to a higher amplitude ratio and to
a smaller phase shift. The second control is catchment subsurface properties, which determine the available storage and how
slowly water leaves the system. The slower the catchment responds, the larger the phase shift and the lower the amplitude
ratio. The Chalk catchments in the UK might be seen as an extreme case where almost all the water infiltrates, and hence the
response time of a single slow reservoir (or perhaps two reservoirs in series) is the main control on the propagation of a periodic
signal. On the other end of the spectrum, there are fully saturated, very responsive catchments mostly along the west coast of
the UK, which behave almost like a single fast reservoir. Using conceptual reservoirs is only one way to interpret the seasonal
signatures. It is useful as many hydrological models are built in that way. There might be, however, other possible ways of
interpretation which we do not consider here.

## 5.3   A hydro-meteorologically more diverse set of catchments – the contiguous US

From Figure 7 it can be seen that for CAMELS catchments (US) the climate indices explain most of the variability in seasonal
response. Again, more humid catchments tend to create more fast flow, and hence they have high amplitude ratios and small
phase shifts. Catchments with a larger moisture index seasonality tend to have smaller phase shifts. In these catchments pre-
cipitation and potential evapotranspiration are mostly out of phase. Therefore, precipitation falls in more humid months, which
might lead to a more flashy response. That means that both precipitation falling on wetter catchments and precipitation falling
in wetter months will be attenuated less. The influence of catchment form is much less pronounced than in the climatically more
homogeneous UK. Continental or global studies tend to identify climate as the dominant hydrological driving force (van Dijk,
2010; Beck et al., 2015), yet regional studies often show other attributes such as geology to be important (for baseflow, see e.g.
Longobardi and Villani, 2008; Bloomfield et al., 2009). Our findings highlight anew that generalising from global to regional
scale, or from regional to global scale, is not straightforward. Such scaling should ideally be done in a process-based way, or
by analysing sub-climates, as the dominance of climate might mask the influence of other factors at large scales. We can also
see that the attribute "fraction of highly productive fractured aquifers" (Figure 6d), which is a hydrogeological classification
available for the UK, shows a much clearer pattern than any soil or geology attributes in the US (see e.g. Figure 7d which
shows the fraction of carbonate sedimentary rock; the same is true if we use e.g. soil permeability for the UK). This might
partly be due to the more heterogeneous US climate which masks the influence of subsurface properties to some degree. But
it might also indicate that the soil or geology data used do not contain the hydrologically relevant soil or geology information.
The hydrogeological classification based on expert judgement available for the UK, even though it is only categorical, might
be more representative of the actual hydro(geo)logical processes at the scale of interest. We therefore cannot conclude that in
the US catchment form does not play a role. We can merely say that the catchment attributes used do not show clear patterns
at the continental scale.





Some of the rather arid catchments in the US plot outside the area that can be modelled by two reservoirs in series or in parallel (Figure 7). This either indicates that we would need another reservoir in series to model the observed phase shift (three reservoirs in series would result in a maximum phase shift of approximately 273 days), that (linear) reservoirs are not a
good description of the hydrological processes, or that the seasonal signature approach breaks down for these arid catchments. Since in water-limited catchments, actual evapotranspiration is typically much smaller than potential evapotranspiration, the input signal we use is very likely a poor proxy for the actual input signal. In very arid catchments ($I_m < -0.5$, dark red dots in Figure 7a), particularly with low moisture seasonality index (Figure 7b), the results should therefore be interpreted with care. It is unclear to what extent these large phase shifts are the result of a poorly approximated input signal or actual
catchment function. This compromises the consistency (McMillan et al., 2017) of the seasonal signatures and makes them most suitable for energy-limited catchments. A way to overcome this limitation would be the use of modelled or measured actual evapotranspiration as input data. As this would require another modelling step or additional data, we leave this for future work (see Supplement for further information).

### 5.4 Can two common hydrological models reproduce the observed seasonal signatures?

The ensemble of IHACRES simulations (Figure 8a) covers the observed range of amplitude ratios and phase shifts (Figure 5a). The BFI pattern also roughly resembles the observed pattern (Figure 6c). Catchments with low BFIs tend to have high amplitude ratios and small phase shifts and vice versa. Figure 8a also shows some patterns that correspond to the theoretical considerations in Section 2 (Figures 2 and 3). To explain this, it is useful to recall the structure of the model. IHACRES consists of a soil moisture deficit store, followed by two parallel linear reservoirs. It thus approximately features the two
examples introduced in Section 2, namely two reservoirs in series or in parallel.

If one of the parallel reservoirs in IHACRES receives very little water (due to an extremely high or low fraction $p$ going into the slow reservoir), the whole system acts like two reservoirs in series. The only difference is that the first reservoir is not a single linear reservoir. It is a non-linear deficit store and thus different from the idealised linear reservoir. This might explain why the upper boundary looks similar to the grey dashed line indicating two linear reservoirs in series, yet not exactly the same.
We did explore how non-linear reservoirs behave in terms of amplitude ratio and phase shift and they seem to behave similar to linear reservoirs (see Supplement). The actual reason for IHACRES not covering the whole area might be the parameters ranges. The parameters ranges used are relatively wide, yet especially the fast reservoir is (to be indeed fast) limited to 5 days, which limits the theoretical space to be smaller than shown in Figure 2.

If the soil moisture reservoir transmits water relatively quickly without much attenuation, the whole system acts like two
reservoirs in parallel. Catchments with similar BFIs have a similar fraction $p$ going into the slow reservoir, which is why we can see patterns (the "yellow stripes") similar to Figure 3a-c (the green, blue, and red stripes). In summary, IHACRES is very similar to the idealised arrangement we introduced in Section 2 and this can be seen in the model output. It would therefore be capable of reproducing the observed seasonal signatures for catchments in the UK and for most of the catchments in the US. Whether IHACRES can reproduce the seasonal signatures, other hydrological signatures and statistical performance metrics
simultaneously is to be explored and beyond the scope of this paper.





The ensemble of GR4J simulations (Figure 8a) only covers a small range of amplitude ratios and phase shifts observed in the UK and the US (Figure 6c). Many of the model runs lead to amplitude ratios higher than unity, which is caused by the groundwater exchange parameter. While this is possible (and can in fact be observed; e.g. in Figure 6c the blue dot outside the grey boundaries is a catchment with water transfer from a neighbouring catchment), it is observed very rarely in the catchments investigated. Furthermore, a non-zero groundwater exchange parameter should ideally be associated with actual water inputs or ouputs (e.g. inter-catchment groundwater flows), and these inputs or outputs are often unknown. No model run leads to a phase shift larger than about 60 days. GR4J also has a soil moisture store followed by two parallel routing stores, i.e. the overall model structure is similar to IHACRES. The stores are, however, not linear reservoirs. They also have fixed parameter values, such as the splitting between fast and slow routing. These internal fixed parameter values might limit the ability of GR4J to reproduce the seasonal signatures. Particularly the flow delay parameter, for which we use a typical parameter range, might be too narrow to produce phase shifts longer than 60 days. From Figure 9 we can see that GR4J tends to produce high BFIs. The internal parameter values, in particular the fixed fraction of 0.9 going through the slow routing store, might limit the signature space of GR4J (note that the BFI clusters around 0.9).

Figure 9 also shows how the BFI and the seasonal signatures vary with different input (forcing). For both models, more humid catchments lead to higher amplitude ratios and smaller phase shifts, and vice versa. This trend, not necessarily the values themselves, agrees with the observed behaviour shown in Figures 6a and 7a.

This analysis is necessarily incomplete for (at least) two reasons. First, we only looked at 40 catchments in the UK to limit the computational demand. More arid catchments (e.g. in the US) might show a different behaviour (e.g. the catchments showing phase shifts larger than 182 days, see Figure 7). Second, the sampling scheme (Latin Hypercube sampling) explores only a subspace of the actual parameter values (both because of the parameter ranges and because of the finite amount of parameter sets). We also made an a-priori decision of how to sample by choosing Latin Hypercube sampling in the first place. This is inevitably subjective, and other sampling schemes might lead to different results. This might especially affect the peaks of the PDFs shown in Figure 9. It is, however, rather unlikely that signature spaces that are highly underrepresented here (e.g. BFIs lower than 0.4 for GR4J) will be covered using different sampling schemes. Wider parameter ranges might change the ranges of the resulting signature spaces. As we use wide ranges based on literature values (see Supplement of Knoben et al., 2019a), our results are at least in line with the practical use of the models, perhaps even more general. This kind of analysis and the seasonal signatures can therefore help to select (or not select) models a-priori, without calibrating them to streamflow data (cf. Vogel and Sankarasubramanian, 2003).

## 6   Conclusions and outlook

We have tested seasonal hydrological signatures aimed at representing how climate seasonality is translated into streamflow seasonality, both approximated by sine waves. The damping (the amplitude ratio) and the phase shift of the incoming sine wave have been used to quantify how catchments respond to seasonal forcing. The presented signatures follow the guidelines of McMillan et al. (2017). The signatures are identifiable, robust, and consistent (see Supplement for further information). They



are representative and have discriminatory power as they exhibit explicable, hydrologically interpretable patterns (Figures 6

and 7), particularly for energy-limited catchments. They can be related to conceptual model structures (arrangements of linear reservoirs, Figure 5), and the model evaluation (Figure 8) has shown that we can indeed observe this theoretical behaviour in model outputs. We have also shown that the signatures can be used as a diagnostic tool since GR4J has been shown to be incapable of modelling the whole observed signature space. As we use precipitation minus potential evapotranspiration as a proxy for the input to a catchment, the seasonal signatures become less reliable for water-limited catchments. To use the

seasonal signatures in water-limited catchments we would need to estimate actual evapotranspiration. The current approach is therefore most suitable for energy-limited catchments, such as catchments in the UK.

We have found that the propagation of the seasonal input through a catchment depends both on climate and catchment form. Climate aridity and seasonality, and corresponding annual and seasonal catchment wetness, drive the partitioning of the incoming forcing into fast and slow flow. Catchment form, such as subsurface properties, influences how strongly the seasonal

input gets attenuated. This is particularly visible in the UK, where the hydrogeological classification available (fraction of highly productive aquifer) can explain the very slow response of some catchments. The more dominant role of climate in the US highlights that scaling from regional to continental (or global) scale is not straightforward and requires thoughtful, ideally process-based approaches. Or in the words of Turner (1989), "conclusions or inferences regarding landscape patterns and processes must be drawn with an acute awareness of scale".

As the seasonal signatures are relatable to conceptual model structures (arrangements of reservoirs), we could also build models based on inference from observed values of the signatures, and not just test existing model structures. This could be done in a stepwise fashion, starting with the seasonal time scale and then adding more complexity if needed (Jothityangkoon et al., 2001; Farmer et al., 2003; McMillan et al., 2011). It would be a step towards model structure identification based on hydrological reasoning, i.e. getting the right answers for the right reasons (Kirchner, 2006). If we decide on a certain model

structure (e.g. two reservoirs in series), we can then use the presented theory to calculate time constants of the reservoirs (the parameters). This might reduce the need for calibration, or at least narrow down the ranges of parameter values. If the time constants obtained from the seasonal signatures differ from time constants obtained by other means, e.g. by calibrating the model using a metric such as KGE, this might be indicative of limitations of typical modelling approaches (Fowler et al., 2018). It might be that the slower annual signal is exciting different parts of the catchments than events (individual peaks or

recessions) do, which we typically calibrate to.

The idea of exploring a model's signature space (following the approach of Vogel and Sankarasubramanian, 2003) perhaps deserves more attention. It allows to explore models systematically and it can reveal whether a model leads to a representative signature space. That is, whether the model has the tendency – or is capable at all – to produce hydrological signature distributions (ranges) that resemble the observed signature distributions (ranges) exhibited by catchment data. Similar to sensitivity

analysis, it allows us to explore and to better understand how a model works, which parameters are important, and what output behaviour a model can generate in general – without calibration. While we limited this analysis to a few signatures, in future studies we should focus on testing whether a model can simultaneously reproduce multiple signatures focusing on different aspects of the hydrological system (Euser et al., 2013; Hrachowitz et al., 2014).





*Code and data availability.* A repository with Matlab code used for the analysis and the resulting data is available from https://github.com/
SebastianGnann/Seasonal_signatures_paper_public. Colours are based on www.ColorBrewer.org, by Cynthia A. Brewer, Penn State. The
MARRMoT toolbox is available from https://github.com/wknoben/MARRMoT. The CAMELS dataset is available from https://ral.ucar.edu/
solutions/products/camels. Information about the UK Benchmark Network can be obtained from https://nrfa.ceh.ac.uk/benchmark-network.
Streamflow data and catchments attributes are available from https://nrfa.ceh.ac.uk. CEH-GEAR precipitation data are available from https:
//doi.org/10.5285/33604ea0-c238-4488-813d-0ad9ab7c51ca. CHESS-PE potential evapotranspiration data are available from https://doi.org/
10.5285/8baf805d-39ce-4dac-b224-c926ada353b7.

*Author contributions.* SJG, NJKH and RAW conceptualised the research project. SJG performed the formal analysis. SJG prepared the
manuscript with contributions from all co-authors.

*Competing interests.* The authors declare that they have no conflict of interest.

*Acknowledgements.* This work is funded as part of the Water Informatics Science and Engineering Centre for Doctoral Training (WISE
CDT) under a grant from the Engineering and Physical Sciences Research Council (EPSRC), grant number EP/L016214/1. Thanks to Wouter
Knoben for help with the MARRMoT toolbox, helpful discussions, and for helpful comments on an earlier version of this manuscript. Thanks
to Gemma Coxon for assisting with the data.





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
