# Peer review of "The response of linear reservoirs to periodic forcing"

_Hydrology and Earth System Sciences, 2019_

## Referee Comment (RC1) · Anonymous Referee #1 · 26 Oct 2019

The focus of this study is on seasonality of forcings (i.e., watershed inputs) and streamflow (i.e., outputs) and how the former is translated into the latter through watersheds functioning. To understand the role of watersheds in dampening of forcings seasonality, authors develop two signatures (namely, the amplitude ratio and the phase shift) and show how combinations of linear models result in certain values for these two signatures. Subsequently, they calculate values for the same signatures using data from several watersheds in the UK and US and overlay the results on top of linear model findings. In this way, they could devise a perceptual model for a given watershed, e.g., two parallel linear reservoirs show to be suitable to model streamflow in some catchment. Finally, authors assess two hydrologic models to figure out whether or not they

could properly reproduce expected variations of these two signatures. This task helps evaluate structural adequacy of a given model. The paper is really well-written, and has high quality presentations. Because this research also provides theoretical foundations for the analyses in this paper, I consider it a great contribution. I believe that the proposed methodology has many applications in the field of watershed modeling and water resources management. Still, I have a few comments that are provided below, which might help improve the quality of this interesting manuscript. I would recommend minor revision.

Comments: Maybe my most major comment is about similarity in concepts between this study and previous studies. Authors themselves also point out that several previous research have essentially relayed the same type of information, but maybe using different techniques (such as unit hydrograph, transit time distributions, etc.). I still do not completely understand what the benefits of the proposed method are, and this requires a dedicated section in the paper. Basically, any other quantitative tools that highlight the differences between the time series characteristics of inputs and outputs could be used here too. For example, we could simply use lag time between forcings and streamflow time series, or maybe variance of these time series, to investigate watershed functioning. For instance, if the ratio between normalized variance of inputs and outputs is really small, watershed might be groundwater dominated. Such a situation would be actually the case with low amplitude ratio under the proposed method. My question is, 'what makes this method unique or better in comparison to other methods? Line 358-359: regarding limitations of this study, authors here mention that "In other climates with a less distinct seasonal pattern, or with two seasons per year our approach will not work". I would argue that there are other limiations that need to be mentioned here too. For example, the proposed method requires quite long records of data. Authors claim that 'inference from observed values of the signatures' is a potential outcome of this method, but as I said, data is needed for this purpose, right? Moreover, most likely the method won't work for sub-annual time scales (because there are lots of hydrological non-linearities at smaller time scales. Maybe, elaborate on different

limitation aspects of this research in a separate section.

Other minor comments: Line 125: explain how multiple linear regression method will be used. I haven't seen any material so far that explains how linear regression could be useful. Line 546: 'reduce the need for calibration'...I don't think so. Maybe, signatures calculated in this research could be used as additional calibration metrics to improve the probability of getting the right answer for the right reasons...but not replacing the calibration process. I have to say that, to me, the most interesting finding in this research is (lines 448-450: the attribute "fraction of highly productive fractured aquifers", which is a hydrogeological classification available for the UK, shows a much clearer pattern than any soil or geology attributes in the US.). This has great applications in model development for ungauged catchments. Minor: Line 16: give a very brief meaning for the word 'seasonality'...later you use terms such as 'mean seasonal regime' or 'seasonal streamflow regime' or 'seasonal signatures', which will make more sense if a clear description of seasonality is provided at the beginning Line 44-45: Shafii and Tolson (2015) is another reference that needs to be cited here Line 73-74: this sentence is a bit unclear: 'a signature describing how climate seasonality is translated into streamflow seasonaltiy adds a timing component with a focus on seasonal and thus slower dynamics.' Line 237: please explain what you mean by 'fast flow routing delay (1 to 5 days)' Thank you

---

## Referee Comment (RC2) · Anonymous Referee #2 · 30 Oct 2019

Referee report on

Hydrological signatures describing the translation of climate seasonality into stream flow seasonality

by

Gnann, Howden and Woods

In their manuscript the authors analyze how long term (seasonal) variations in precipitation time series translate into (long term) variations in stream flow. To do so the authors decompose the precipitation and corresponding stream flow time series into

[Figure]

**HESSD**

their Fourier modes and analyze the mode corresponding to the annual (seasonal) cycle.

The paper is well written and addresses the problem of signal and forcing from a point of view which is more common in electro-technical engineering than in hydrology. Thus the paper may help to stimulate the field by introducing new methods and alternative approaches to analyze the relation between input-output time series.

Below some comments and suggestions which should help the authors to improve and strengthen their manuscript. Abstract: "We approximate [..] by sine waves." Input and output signals are not periodic per se, but show recurring patterns. In order to address this point the authors may simply rephrase the above statement with something like: "In order to analyze the seasonality relations between input [...] and output we represent the two time series by their seasonal (annual) Fourier mode."

Such a formulation avoids the criticism that the signal itself periodic, while keeping all the rest of the analysis unchanged.

Sec 2.2.1: 1 year Fourier mode: It would be interesting to see for an example how the different Fourier modes are represented in the spectrum of the time series. Such a measure would show how "strong" the annual mode is compared to the other modes of the signal.

Line 110: Although notation is an arbitrary choice, I would suggest the authors to use "PET" or at least "E_{PET}" in order to refer to Potential Evapo-Transpiration. Reducing the in-/output signal by putting all weight of the time-series into the single (seasonal) Fourier mode may be problematic for analyzing real world data where: a. It is not per se clear that the overall dominant part of the signal. (Here as mentioned above the spectrum should give insight)

b. Additionally the different modes of the input signal do not necessarily need to be linearly coupled with modes of the same frequency in the output.

Thus, it should be made clear that the description in section 2.1 relies on the assumption of a single wavelength forcing and a linear response system.

Note: Due to linearity, all derivations presented in 2.1 should be valid for any Fourier component of the forcing function with $F_n = A_n \exp(i*k*t)$ where $A_n$ is the amplitude of the corresponding mode in the Fourier series.

Figure 4: As mentioned before it would be interesting to see, how the blue and orange modes are represented in the corresponding spectra. If the seasonal modes are by far the most dominant frequencies in the signal it could help to justify for the single mode forcing model.

Sec. 4.2: Given the heterogeinity of natural systems it is not too surprising that a single linear (reservoir) model is not sufficient.

Fig. 6a and 7a: I would suggest the authors to use a two color divergent color scale to distinguish between negative and positive $l\_m$ (blue to white for neg. and white to red for pos values)

Another critique of Figs.6/7 is that the high point density can hide variabilities, especially when the points are plotted in a sorted manner, e.g. sorted by amplitude In order to avoid such a situation one could first randomize the sample with respect to the variable of the color bar.

Section 4.3 requires some more details how the models were set up and parameters were varied/chosen (This can be added to the SI).

Examples are:

Line 333: Running IHACRES with 20 000 parameter sets. - Which are the parameters? - What are the parameter ranges that were varied?

Line 335: The sentence "Plotting curves [...] produced by a certain set ..." needs some clarification. Questions which may arise here are: - How was the parameter set being

chosen? - Was it always the same for all different catchments? - Did the authors perform a parameter sensitivity analysis?

Line 343: "[...] with varying forcing.": Why do the authors introduce here the aridity index $AI=PET/P=1-F/P$ as a nonlinear transformed quantity of $F=P-PET$ rather than using their definition directly.

Alternatively if the aridity dependence is the point to make here the authors should simply say this: "[...] does not vary substantially with varying $AI=PET/P=1-F/P$."

— I hope that the authors find my comments & suggestions useful to to improve the manuscript and strengthen their arguments.

---

## Author Comment (AC1) · 8 Nov 2019

**Response to Referee #1**

Responses are written in blue.

The focus of this study is on seasonality of forcings (i.e., watershed inputs) and streamflow (i.e., outputs) and how the former is translated into the latter through watersheds functioning. To understand the role of watersheds in dampening of forcings seasonality, authors develop two signatures (namely, the amplitude ratio and the phase shift) and show how combinations of linear models result in certain values for these two signatures. Subsequently, they calculate values for the same signatures using data from several watersheds in the UK and US and overlay the results on top of linear model findings. In this way, they could devise a perceptual model for a given watershed, e.g., two parallel linear reservoirs show to be suitable to model streamflow in some catchment. Finally, authors assess two hydrologic models to figure out whether or not they could properly reproduce expected variations of these two signatures. This task helps evaluate structural adequacy of a given model. The paper is really well-written, and has high quality presentations. Because this research also provides theoretical foundations for the analyses in this paper, I consider it a great contribution. I believe that the proposed methodology has many applications in the field of watershed modeling and water resources management. Still, I have a few comments that are provided below, which might help improve the quality of this interesting manuscript. I would recommend minor revision.

We thank reviewer #1 for the helpful and encouraging feedback.

Comments: Maybe my most major comment is about similarity in concepts between this study and previous studies. Authors themselves also point out that several previous research have essentially relayed the same type of information, but maybe using different techniques (such as unit hydrograph, transit time distributions, etc.). I still do not completely understand what the benefits of the proposed method are, and this requires a dedicated section in the paper. Basically, any other quantitative tools that highlight the differences between the time series characteristics of inputs and outputs could be used here too. For example, we could simply use lag time between forcings and streamflow time series, or maybe variance of these time series, to investigate watershed functioning. For instance, if the ratio between normalized variance of inputs and outputs is really small, watershed might be groundwater dominated. Such a situation would be actually the case with low amplitude ratio under the proposed method. My question is, 'what makes this method unique or better in comparison to other methods?

Thank you for pointing that out. We have indeed pointed out similarities to other techniques, we however think that they do not necessarily relate to the same type of information. Transit times focus on the velocity of water particles and therefore yield different insights. Many other methods (unit hydrograph, lag time, variance of time series) focus on shorter time scales. We believe that the focus on seasonal dynamics can yield related yet additional information compared to methods focusing on event scales. Furthermore, we chose the approach because there are analytical solutions for how sine waves are propagated by linear systems. This allows for example to interpret the results in terms of configurations of linear reservoirs and to estimate their associated time constants. The suggested ratio of normalised variances will probably be related to the seasonal signatures, yet how exactly can such a number be interpreted beyond a qualitative statement like "this watershed might be groundwater dominated"? We will clarify the motivation for our approach in a revised manuscript.

Line 358-359: regarding limitations of this study, authors here mention that "In other climates with a less distinct seasonal pattern, or with two seasons per year our approach will not work". I would argue that there are other limiations that need to be mentioned here too. For example, the proposed method requires quite long records of data.

From the SI it can be seen that 10 years are enough to obtain a robust result for most places. But of course, we require at least a couple of years (i.e. seasonal cycles) to meaningfully fit a sine curve. We will add a sentence about data limitations.

Authors claim that 'inference from observed values of the signatures' is a potential outcome of this method, but as I said, data is needed for this purpose, right?

The reviewer is correct that data is required for this purpose. We will clarify the sentence to make it clearer regarding what can be inferred from the signatures.

Moreover, most likely the method won't work for sub-annual time scales (because there are lots of hydrological non-linearities at smaller time scales.

We agree with the reviewer here. We decided to focus on the annual time scale because it has a clear physical meaning (see lines 106-110) and because the seasonal flow regime is of importance to many applications. We will emphasise that in a revised manuscript.

Note that SI 1.4 briefly investigates non-linear reservoirs.

Maybe, elaborate on different limitation aspects of this research in a separate section.

We will add a discussion of the limitations you mentioned to Section 5.1 and change the title of that section. We think that another separate section on limitations might not necessarily be helpful. For example, we discuss the limitations of the modelling exercise in Section 5.4 (line 507-519), where we think it fits best.

Other minor comments: Line 125: explain how multiple linear regression method will be used. I haven't seen any material so far that explains how linear regression could be useful.

We used multiple linear regression to fit sine waves to data. This is explained in SI 2.1.2. We will add a clearer reference to that in the text.

Line 546: 'reduce the need for calibration'. . .I don't think so. Maybe, signatures calculated in this research could be used as additional calibration metrics to improve the probability of getting the right answer for the right reasons. . .but not replacing the calibration process.

Once a certain arrangement of linear reservoirs is chosen, the signatures are associated with time constants of these reservoirs. For example, if we chose a model consisting of two reservoirs in series, the theory can be used to obtain the two time constants of the reservoirs. This might not replace the calibration process completely, but it could be used to limit parameter ranges or to fix certain parameters. Since we haven't tested that yet, we can't say whether that will be useful in practice. Yet in any case, as you have said, the signatures might be used as an additional calibration metric (which is also indicated by our modelling experiment). We will revise the paragraph to clarify this.

I have to say that, to me, the most interesting finding in this research is (lines 448-450: the attribute "fraction of highly productive fractured aquifers", which is a hydrogeological classification available for the UK, shows a much clearer pattern than any soil or geology attributes in the US.). This has great applications in model development for ungauged catchments.

Thank you. The question remains of how to get such a classification for other places than the UK.

Minor: Line 16: give a very brief meaning for the word 'seasonality'. . .later you use terms such as 'mean seasonal regime' or 'seasonal streamflow regime' or 'seasonal signatures', which will make more sense if a clear description of seasonality is provided at the beginning

We will revise the first paragraph to clarify the meaning of the word seasonality.

Line 44-45: Shafii and Tolson (2015) is another reference that needs to be cited here

We will add that reference.

Line 73-74: this sentence is a bit unclear: 'a signature describing how climate seasonality is translated into streamflow seasonaltiy adds a timing component with a focus on seasonal and thus slower dynamics.'

The obtained phase shift tells us how long – on average – the seasonal forcing peak is delayed before it becomes the seasonal streamflow peak. This time lag (e.g. 1 month) is what we mean by timing component. We will revise that sentence.

Line 237: please explain what you mean by 'fast flow routing delay (1 to 5 days)'

We will add a more detailed description of the model parameters in the SI.

Thank you

Thank you for your review!

---

## Author Comment (AC2) · 8 Nov 2019

**Response to Referee #2**

Responses are written in blue.

Referee report on Hydrological signatures describing the translation of climate seasonality into stream flow seasonality by Gnann, Howden and Woods

In their manuscript the authors analyze how long term (seasonal) variations in precipitation time series translate into (long term) variations in stream flow. To do so the authors decompose the precipitation and corresponding stream flow time series into their Fourier modes and analyze the mode corresponding to the annual (seasonal) cycle.

The paper is well written and addresses the problem of signal and forcing from a point of view which is more common in electro-technical engineering than in hydrology. Thus the paper may help to stimulate the field by introducing new methods and alternative approaches to analyze the relation between input-output time series. Below some comments and suggestions which should help the authors to improve and strengthen their manuscript.

We thank reviewer #2 for the helpful and encouraging feedback.

Abstract: "We approximate [..] by sine waves." Input and output signals are not periodic per se, but show recurring patterns. In order to address this point the authors may simply rephrase the above statement with something like: "In order to analyze the seasonality relations between input [...] and output we represent the two time series by their seasonal (annual) Fourier mode." Such a formulation avoids the criticism that the signal itself periodic, while keeping all the rest of the analysis unchanged.

Thank you for the suggestion, we will revise the text accordingly.

Sec 2.2.1: 1 year Fourier mode: It would be interesting to see for an example how the different Fourier modes are represented in the spectrum of the time series. Such a measure would show how "strong" the annual mode is compared to the other modes of the signal.

We did a quick analysis to check how strong the annual mode is in comparison to other modes. We calculated one-sided power spectra and extracted their maxima for all catchments. Two examples (following a copy of Figure 4 from the paper) are shown below (Fig. 2).

[Figure]

*Fig 1.* Climate input (*P* - *E_p*) and catchment output (*Q*) for two catchments in the UK, and their respective seasonal components. The time series are smoothed using a 30-day moving mean. The Ericht is a rather responsive catchment (BFI =

0.47), while the East Avon has a large baseflow component (BFI = 0.89). Note that for the bottom plots ("Seasonal") the mean values of the sine curves are set to zero. (Figure 4 in the manuscript.)

[Figure]

*Fig 2.* One-sided power spectra of climate input ($P$ - $E_p$; blue) and catchment output ($Q$; orange) for two catchments in the UK.

For almost every catchment in our manuscript (~99%) the strongest forcing Fourier mode is the annual mode. For a few catchments in the US a 0.5y mode is the strongest, yet typically there is also a 1y mode present. Some of the streamflow data show strongest modes different from 1. Yet again, this doesn't mean that there is no annual mode present. For example, in panel (b) below we can see a strong multi-annual mode and the annual mode. We can also see that the groundwater dominated catchment (b) seems to act as a low-pass filter, dampening signals with shorter periods stronger than signals with longer periods. In principle, we could analyse more periods than the annual period and perhaps contrasting different periods might yield other interesting insights. But we have decided to focus on the annual time scale because it has a clear physical meaning (see lines 106-110). We will add the Fourier spectra to the SI.

Line 110: Although notation is an arbitrary choice, I would suggest the authors to use "PET" or at least "E_{PET}" in order to refer to Potential Evapo-Transpiration.

Thank you for the suggestion, but we would prefer to stick with our notation.

Reducing the in-/output signal by putting all weight of the time-series into the single (seasonal) Fourier mode may be problematic for analyzing real world data where: a. It is not per se clear that the overall dominant part of the signal. (Here as mentioned above the spectrum should give insight)

See above for an answer to that question and for Fourier spectra.

b. Additionally the different modes of the input signal do not necessarily need to be linearly coupled with modes of the same frequency in the output. Thus, it should be made clear that the description in section 2.1 relies on the assumption of a single wavelength forcing and a linear response system.

We will state these assumptions more clearly in a revised form of the manuscript.

Note: Due to linearity, all derivations presented in 2.1 should be valid for any Fourier component of the forcing function with F_n=A_n\exp(i*k*t) where A_n is the amplitude of the corresponding mode in the Fourier series.

Yes, the theory is not limited to the annual model. Yet as we've noted above, we focus on the annual mode as it is the dominant mode and as it has a clear physical driving force.

Figure 4: As mentioned before it would be interesting to see, how the blue and orange modes are represented in the corresponding spectra. If the seasonal modes are by far the most dominant frequencies in the signal it could help to justify for the single mode forcing model.

See above for an answer to that question and for Fourier spectra.

Sec. 4.2: Given the heterogeinity of natural systems it is not too surprising that a single linear (reservoir) model is not sufficient.

We agree on that, but we thought we start with rejecting the simplest model.

Fig. 6a and 7a: I would suggest the authors to use a two color divergent color scale to distinguish between negative and positive l_m (blue to white for neg. and white to red for pos values)

We originally intended to stick with the RGB colour schemes introduced by Knoben et al., 2018. We agree, however, that the colour scale is not the best choice in our case. We will change that accordingly.

Another critique of Figs.6/7 is that the high point density can hide variabilities, especially when the points are plotted in a sorted manner, e.g. sorted by amplitude In order to avoid such a situation one could first randomize the sample with respect to the variable of the color bar.

At the moment, the points are plotted based on the list of catchments we've used. That is, neither completely random (the catchment list tends to follow geographical locations) nor sorted by anything specific such as amplitude ratio. We will check whether the plotting order influences the figure and improve the information content of the plot if possible.

Section 4.3 requires some more details how the models were set up and parameters were varied/chosen (This can be added to the SI). Examples are: Line 333: Running IHACRES with 20 000 parameter sets. - Which are the parameters? - What are the parameter ranges that were varied? Line 335: The sentence "Plotting curves [...] produced by a certain set ..." needs some clarification. Questions which may arise here are: - How was the parameter set being chosen? - Was it always the same for all different catchments? - Did the authors perform a parameter sensitivity analysis?

Thanks for pointing out places where we were unclear in the modelling part. We will add more details on the modelling part to the SI.

Line 343: "[...] with varying forcing.": Why do the authors introduce here the aridity index AI=PET/P=1-F/P as a nonlinear transformed quantity of F=P-PET rather than using their definition directly. Alternatively if the aridity dependence is the point to make here the authors should simply say this: "[...] does not vary substantially with varying AI=PET/P=1-F/P."

Thanks for pointing that out. Indeed, using the aridity index here is not necessary. The main purpose was to point out that each line corresponds to a different forcing input. We will change that to the moisture index $I_m$ so that it's consistent with Figures 6 & 7.

— I hope that the authors find my comments & suggestions useful to to improve the manuscript and strengthen their arguments.

Thanks again for reviewing our manuscript!

References

Knoben, W.J., Woods, R.A. and Freer, J.E., 2018. A Quantitative Hydrological Climate Classification Evaluated With Independent Streamflow Data. Water Resources Research, 54(7), pp.5088-5109.

---

## Short Comment (SC1) · 10 Nov 2019

This review was prepared as part of graduate program course work at Wageningen University, and has been produced under supervision of dr Ryan Teuling by a student that prefers to stay anonymous. The review has been posted because of its good quality, and likely usefulness to the authors and editor. This review was not solicited by the journal.

Peer review on "Hydrological signatures describing the translation of climate seasonality into streamflow seasonality" by Gnann et al.

[Figure]

The manuscript "Hydrological signatures describing the translation of climate seasonality into streamflow seasonality" by Gnann et al. proposes two new hydrological signatures: the amplitude ratio and phase shift between the climatic forcing and the streamflow. The aim of this research is to use these signatures to quantify the catchment response to climatic forcing and use them for model evaluation. To determine the amplitude ratio and the phase shift, a sine function is fitted through both the climatic forcing and the streamflow. The climatic forcing is defined as the precipitation minus the potential evapotranspiration. The signatures are interpreted with the response (signatures) of linear reservoirs in series or parallel to climatic forcing. To test if the signature values are hydrologically interpretable, signatures for catchments in the UK and the US are defined and related to catchment characteristics to see if there is a pattern. Two models are discussed based on the signature range that they can produce. The authors conclude that the signatures can be used for model evaluation and to help model builders decide on the model configuration. The use of hydrological signatures to define a model configuration is a novelty, it would be interesting to look for other hydrological signatures and further investigate the abilities of this method. The phase shift is an interesting signature because it could quantify the time delay between climatic forcing and streamflow. However, my main concern is on the way the signatures are used here to evaluate models. The method is not appropriate, the model evaluation is not complete and no comparison is made with other evaluation methods. Furthermore, I also have some critical remarks on the proposed new signatures. They have a low accuracy and are not widely applicable. My last concern is about the conclusions, which are all based on visual interpretation instead of statistical analysis. Because of these reasons, I do not see the added value of this manuscript to the existing body of literature and therefore I recommend to reject the manuscript.

To start with, I will explain my main concern on the model evaluation using the proposed method. In the paper a new way of model evaluation is proposed, namely looking at range of values of signatures (phase shift and amplitude ratio) that different models can produce. To test how large the range of produced signatures by the models is, a Monte-

[Figure]

Carlo sampling experiment is done. The authors state that this new method could be more meaningful and fit-for-purpose than already existing model evaluation methods: "Signatures rooted in hydrological theory offer a potentially more meaningful and fit-for-purpose alternative to the typically used statistical metrics such as the Nash-Sutcliffe efficiency (NSE; Nash and Sutcliffe". I do not agree with this argument, I will discuss the flaws of this method in the next paragraph. First, I would like to raise attention to the fact that only two models are tested and no comparison is made with already existing model evaluation methods. I think a much more extensive approach is needed if they want to propose this as an alternative for the already existing model evaluation methods. More models need to be tested and the outcome of this evaluation method needs to be compared with outcomes of other model evaluation methods to see if they are in line and whether this method really gives more meaningful outcomes.

First of all, in the manuscript only two figures show the results of the model evaluation with this new method. These figures alone, are not enough to evaluate the two models. Quantitative statements on the model functioning are needed, i.e. how well does the model predict the streamflow? All conclusions are based on visual interpretation, but graphs can sometimes be misleading, statistical analysis would be much more appropriate to compare different models on their functioning. In this manuscript only two models are tested, but if a lot of models need to be tested, numbers would make it easier to tell which model is best instead of comparing a lot of graphs. Secondly, the choice of catchments used for model evaluation influences the outcome. For this experiment, 40 catchments in the UK are used. However, the UK catchments show better relationships between the signatures and catchment relationships (see figure 6 & 7), so the choice of using UK catchments instead of US catchments influences the outcome of testing this method. Thirdly, the number of parameters differs for the two models. Whether the difference between the signature space of the models is due to model functioning instead of the used range for different parameters, is questionable. My suspicion increases when reading line 481-483 "The actual reason...in Figure 2." and line 500-501 "Particularly the flow...than 60 days.", it seems that the signature output is determined by the parameter range instead of the model functioning, so how will this method evaluate models in an objective way then? The conclusion that the signatures are a diagnostic tool because GR4J is not capable of modelling the whole signature space (Line 427-428), is thus not valid in my opinion!

Lastly, only a small part of model predictability aspects is evaluated. Pechlivanidis et al. (2011) summarized different model evaluation methods, where they discuss different objective approaches. Objective functions are here defined as numerical measures of the difference between the model simulated output and the observed (measured) catchment output. The Nash-Sutcliffe Efficiency (NSE) and Kling and Gupta Efficiency (KGE) are examples of this approach. The proposed method here is an objective function as well, since produced signatures by models are compared with observed signatures of catchments. The KGE has been introduced to overcome some limitations of NSE, this method analyses the correlation, the bias, and a measure of relative variability in the simulated and observed values (Gupta et al., 2009). This method evaluates thus more aspects of model functioning than the new proposed method here, which only gives an indication of the ability of a model to attenuate the climate forcing into a streamflow signal with a right time delay (if the signatures are correct!), but not if the model can produce the right streamflow variability and mean, peak and low flows.

The authors could improve the method by evaluating the models based on more hydrological signatures and quantify the model functioning. Furthermore, they could do test more models and compare the outcomes with other model evaluation methods. They could also improve the transparency of this method by adding a table with the changed parameters and the range. Finally, they should argue why although different parameters of the two models are changed, the model outcomes can still be compared.

My second major concern is about the signatures, they have a low accuracy and are not widely applicable. First, I will address the accuracy of the signatures. To determine the phase shift and amplitude ratio, a sine function is fitted on the climate forcing (P-ETp) and the streamflow. The method of fitting a sine function through the forcing and

streamflow time series does not seem adequate to me. Most of the catchment regimes do not show a clear sinusoidal yearly cycle. This is well visible in the 16 different regime types, after Weingartner and Aschwanden (1992). For example, catchments that show two discharge peaks in one year cannot be described well by a sine function, this will lead to an error in the phase shift. In the paper two examples are given were a sinusoidal function is fitted on the climate forcing and streamflow. The timing of the sine function (phase) on the forcing and streamflow seems to be quite good in these cases. However, the sine function does not follow peak discharge and low discharges. This is clearly visible in the middle figure of the East Avon at Upavon catchment, the discharge peaks in 2001 and 2003 are not represented in the sine function (discharge is double the fitted discharge!), also the sine function does not follow the discharge in 2005/2006 when there is a low discharge. This shows that the sine fitting leads to errors in the amplitude ratio. Since the signatures are used for model evaluation, these errors could also lead to errors in the outcome of the model evaluation.

Furthermore, the use of the potential evapotranspiration (ETp) leads to errors (and thus lower accuracy) in the signatures for semi-arid and arid catchments, since the potential evapotranspiration deviates from the actual evapotranspiration in these areas. This problem is mentioned by the authors in line 380-385. This problem can be solved by including a model to estimate the actual evapotranspiration (also mentioned by the authors). This would also help interpreting the signatures with the catchment characteristics. For example, in line 436-438 the authors state that the signatures of the US catchments show a relation with the moisture index. This conclusion is made based on visual interpretation of figure 7a. But I think this conclusion is not valid because the signatures of the dry catchments on the left side have a large uncertainty because of the use of ETp instead of ETact.

The other disadvantage of the signatures is that they are not widely applicable. The problem of using the signatures for arid and semi-arid areas is already mentioned, but this could be solved by using the actual evapotranspiration. However, these signatures

are also not valid for catchments with precipitation falling as snow. Since catchments with precipitation as snow show a typically seasonal cycle, the need of leaving these out of consideration is a major lack of the proposed signatures. Furthermore, the signatures are also not valid for climates with a less distinct seasonal pattern, so this will further limit the applicability of the signatures. Because the signatures can only be used for a certain type of catchments, it is the question whether they contain new information on the streamflow seasonality of these catchments. There are already hydrological signatures that describe the response of streamflow to climatic forcing, for example the flow duration curve. A steep slope in the flow duration curve indicates a fast response of the streamflow to climate forcing whereas a flatter curve indicates a relatively damped response and higher storage (Yadav, 2007). Only the timing component might add new information, but since the method of determining the phase shift is not accurate, I do not see the added value.

The last thing I would like to point out is that all conclusions based on visual interpretation instead of statistical analysis. For the sine fitting method, I would like to see the goodness of it or the sum of squared errors (SS), to know how well the fit of the sine function to the climatic forcing and streamflow is. For the relationships between the signatures and catchment characteristics, it would be better to calculate the correlation coefficient instead of only the visual interpretation, since this might be misleading. The same goes for the model evaluation method, it would be nice to have a quantitative statement on how well the model works. This would also make it easier to compare more models, as mentioned before.

Minor issues and typo's:

Minor issue 1: Line 68-70 "All of these ... streamflow seasonality." I am not convinced. For example the slope of the flow duration curve can say something about the translation of climatic forcing into streamflow seasonality. A steep slope in the flow duration curve indicates a fast response of the streamflow to precipitation inputs whereas a flatter curve indicates a relatively damped response and higher storage (Yadav, 2007).

Minor issue 2: Line 94-96 "The amplitude might ... seasonal component alone." Why stating this if it is not done for this research, is it a follow up research suggestion? Then it should be placed in the discussion.

Minor issue 3: "catchment form" can better be replaced by catchment characteristics (For example in line 100). Catchment form suggest you are looking at the effect of a small river with a lot of branches or a stretched river.

Minor issue 4: The aim could be stated much more clearly, "test whether the seasonal signatures are useful for modelling practice (line 101)" not specific enough.

Minor issue 5: Line 110/111 "We use Ep ... would be needed." Not a valid argument, how much would the uncertainty increase if you add another model?

Minor issue 6: In line 124 a small remark is made on the method of the sine fitting. This could be elaborated a bit more. Why use the sine fitting method? Which methods did you compare and why did you choose for the linear regression method (it is now in the supplement, but I think it is better to include it in the text)?

Minor issue 7: A reference is needed to support line 200 "The upper limit...shape parameter equal to 2." Minor issue 8: About figure 3, could you explain the form of the curve when ?2 becomes larger and ?1 and fraction going to second reservoir are constant.

Minor issue 9: Line 235, explain the choice for Latin Hypercube sampling.

Minor issue 10: Table 1, add more information on range variables. For example for moisture index: -1= most arid and 1= most humid.

Minor issue 11: Figure 4: add color indication to description, climatic forcing (blue) and streamflow (orange).

Minor issue 12: Figure 5: Based on what criteria are the benchmark catchments chosen (grey dots)? Same goes for the two red dots, random or do they represent a certain

type of catchments?

Minor issue 13: Line 284, missing reference to table 1. Catchment attributes

Minor issue 14: Line 300, missing reference to table 1. Catchment attributes

Minor issue 15: Line 304-305 "Yet generally, ...in figure 6)." Statement is not explained in discussion, why are the US phase shift larger than for the UK catchments?

Minor issue 16: Figure 9b, Higher probability for high BFI for GR4J than IHACRES, but GR4J lower phase shift (max 60 days)!! Why? I would expect a larger phase shift when a larger part of the flow is slow flow.

Minor issue 17: Line 393-395 "Since the BFI... seasonal signatures." I do not agree, the BFI cannot be used as a cause for observed patterns, but it can be related to the observed pattern. A higher base flow means more slow flow so this could be related to a larger phase shift.

Typo's:

Line 17: sensitive Line 64: minimum Line 73: seasonality Line 278: reproduce Line 496: outputs

References:

Gupta, H. V., Kling, H., Yilmaz, K. K., and Martinez, G. F. (2009). Decomposition of the mean squared error and NSE performance criteria: Implications for improving hydrological modelling, Journal of Hydrology, 377, 80–91

Pechlivanidis, I. G., Jackson, B. M., McIntyre, N. R., & Wheater, H. S. (2011). Catchment scale hydrological modelling: a review of model types, calibration approaches and uncertainty analysis methods in the context of recent developments in technology and applications. Global NEST journal, 13(3), 193-214.

Weingartner, R., & Aschwanden, H. (1992). Discharge regime–the basis for the estimation of average flows. Hydrological Atlas of Switzerland, Plate, 5, 26.

Yadav, M., Wagener, T., and Gupta, H. (2007). Regionalization of constraints on expected watershed response behavior for improved predictions in ungauged basins, Advanced Water Resources, 30, 1756–1774

---

## Author Comment (AC3) · 22 Nov 2019

**Response to Referee #3**

Responses are written in blue.

This review was prepared as part of graduate program course work at Wageningen University, and has been produced under supervision of dr Ryan Teuling by a student that prefers to stay anonymous. The review has been posted because of its good quality, and likely usefulness to the authors and editor. This review was not solicited by the journal.

Peer review on "Hydrological signatures describing the translation of climate seasonality into streamflow seasonality" by Gnann et al.

The manuscript "Hydrological signatures describing the translation of climate seasonality into streamflow seasonality" by Gnann et al. proposes two new hydrological signatures: the amplitude ratio and phase shift between the climatic forcing and the streamflow. The aim of this research is to use these signatures to quantify the catchment response to climatic forcing and use them for model evaluation. To determine the amplitude ratio and the phase shift, a sine function is fitted through both the climatic forcing and the streamflow. The climatic forcing is defined as the precipitation minus the potential evapotranspiration. The signatures are interpreted with the response (signatures) of linear reservoirs in series or parallel to climatic forcing. To test if the signature values are hydrologically interpretable, signatures for catchments in the UK and the US are defined and related to catchment characteristics to see if there is a pattern. Two models are discussed based on the signature range that they can produce. The authors conclude that the signatures can be used for model evaluation and to help model builders decide on the model configuration. The use of hydrological signatures to define a model configuration is a novelty, it would be interesting to look for other hydrological signatures and further investigate the abilities of this method. The phase shift is an interesting signature because it could quantify the time delay between climatic forcing and streamflow. However, my main concern is on the way the signatures are used here to evaluate models. The method is not appropriate, the model evaluation is not complete and no comparison is made with other evaluation methods. Furthermore, I also have some critical remarks on the proposed new signatures. They have a low accuracy and are not widely applicable. My last concern is about the conclusions, which are all based on visual interpretation instead of statistical analysis. Because of these reasons, I do not see the added value of this manuscript to the existing body of literature and therefore I recommend to reject the manuscript.

Thank you for your review and the feedback on our work.

To start with, I will explain my main concern on the model evaluation using the proposed method. In the paper a new way of model evaluation is proposed, namely looking at range of values of signatures (phase shift and amplitude ratio) that different models can produce. To test how large the range of produced signatures by the models is, a Monte Carlo sampling experiment is done. The authors state that this new method could be more meaningful and fit-for-purpose than already existing model evaluation methods: "Signatures rooted in hydrological theory offer a potentially more meaningful and fit-for purpose alternative to the typically used statistical metrics such as the Nash-Sutcliffe efficiency (NSE; Nash and Sutcliffe". I do not agree with this argument, I will discuss the flaws of this method in the next paragraph.

The reviewer introduces a concern about the use of hydrological signatures for model evaluation, especially in exploring the range of hydrological responses that a model can produce. We do not make any claims about the novelty of the evaluation method itself. As stated in the manuscript, the

idea of evaluating a model's response before calibration follows the idea of Vogel and Sankarasubramanian (2003).

First, I would like to raise attention to the fact that only two models are tested and no comparison is made with already existing model evaluation methods. I think a much more extensive approach is needed if they want to propose this as an alternative for the already existing model evaluation methods. More models need to be tested and the outcome of this evaluation method needs to be compared with outcomes of other model evaluation methods to see if they are in line and whether this method really gives more meaningful outcomes.

We would like to emphasise that the model evaluation is not the primary point of this paper, but the presentation of the seasonal signatures, which is one reason why we've kept the modelling part reasonably short. We will try to emphasise that more clearly in a revised version the manuscript.

We do not intend to present a full alternative to existing model evaluation methods. We primarily want to show how the signatures might be used as an additional source of information in model evaluation. We agree that a more extensive approach would be needed if the aim was a comparison to existing model evaluation methods, but this is not our intention.

First of all, in the manuscript only two figures show the results of the model evaluation with this new method. These figures alone, are not enough to evaluate the two models. Quantitative statements on the model functioning are needed, i.e. how well does the model predict the streamflow? All conclusions are based on visual interpretation, but graphs can sometimes be misleading, statistical analysis would be much more appropriate to compare different models on their functioning. In this manuscript only two models are tested, but if a lot of models need to be tested, numbers would make it easier to tell which model is best instead of comparing a lot of graphs.

We do not aim at evaluating whether streamflow is predicted well or not. In fact, we wouldn't expect streamflow to be predicted well based on the seasonal signatures alone, since they only aim at a certain aspect of the catchment response. The aim here is not to compare model runs from individual parameter sets with observed streamflow. We are primarily interested in the overall capabilities of the models. From Figure 6 we can see that GR4J (given the parameter ranges used) cannot reproduce what we observe. The question we try to answer here is not "which model is best". Rather, we want to test whether a certain model (given the parameter ranges used) is generally capable of producing the range of observed signatures, and thus cannot be rejected (see Vogel and Sankarasubramanian, 2003).

Secondly, the choice of catchments used for model evaluation influences the outcome. For this experiment, 40 catchments in the UK are used. However, the UK catchments show better relationships between the signatures and catchment relationships (see figure 6 & 7), so the choice of using UK catchments instead of US catchments influences the outcome of testing this method.

We agree that the choice of catchments influences the outcome. But we think that choosing catchments in the UK is reasonable exactly because the seasonal signatures we propose are more robust in the mostly energy-limited UK. We will state our reasoning for using this subset of catchments more clearly in a revised version of the manuscript. We will also emphasise that the results of the modelling experiment are only valid for the UK.

Thirdly, the number of parameters differs for the two models. Whether the difference between the signature space of the models is due to model functioning instead of the used range for different parameters, is questionable. My suspicion increases when reading line 481-483 "The actual

reason...in Figure 2." and line 500-501 "Particularly the flow...than 60 days.", it seems that the signature output is determined by the parameter range instead of the model functioning, so how will this method evaluate models in an objective way then? The conclusion that the signatures are a diagnostic tool because GR4J is not capable of modelling the whole signature space (Line 427-428), is thus not valid in my opinion!

We agree that the results depend on the parameter ranges. Specifying parameter ranges always involves some subjective judgment. We mostly used the default ranges from the MARRMoT toolbox (Knoben et al., 2019), which are intended to be wide. We will have a look at recent literature on the parameter ranges. We will investigate whether broader ranges influence the results and we will update the parameter ranges if necessary. We will also try to emphasise the limitations of choosing certain parameter ranges more clearly.

Lastly, only a small part of model predictability aspects is evaluated. Pechlivanidis et al. (2011) summarized different model evaluation methods, where they discuss different objective approaches. Objective functions are here defined as numerical measures of the difference between the model simulated output and the observed (measured) catchment output. The Nash-Sutcliffe Efficiency (NSE) and Kling and Gupta Efficiency (KGE) are examples of this approach. The proposed method here is an objective function as well, since produced signatures by models are compared with observed signatures of catchments. The KGE has been introduced to overcome some limitations of NSE, this method analyses the correlation, the bias, and a measure of relative variability in the simulated and observed values (Gupta et al., 2009). This method evaluates thus more aspects of model functioning than the new proposed method here, which only gives an indication of the ability of a model to attenuate the climate forcing into a streamflow signal with a right time delay (if the signatures are correct!), but not if the model can produce the right streamflow variability and mean, peak and low flows. The authors could improve the method by evaluating the models based on more hydrological signatures and quantify the model functioning. Furthermore, they could do test more models and compare the outcomes with other model evaluation methods.

We absolutely agree that in a "general" model evaluation, we should look at other aspects of the hydrograph, ideally by using multiple hydrologically interpretable signatures. Yet we did not intend to evaluate these two models in general, but only with respect to the proposed signatures. We also agree that the proposed signatures could be used as an objective function, we however decided against such an evaluation approach. Instead, we focused on the range of possible model responses. This doesn't mean that for individual catchments, the model would have to be rejected, but as a model for all the catchments investigated, it would have to be rejected (given the parameter ranges chosen). This might be particularly helpful for large sample studies, where often one or a few model structures are chosen a-priori for all catchments (see also Addor and Melsen, 2019, who show how models are often chosen based on legacy rather than adequacy) .

They could also improve the transparency of this method by adding a table with the changed parameters and the range.

Thank you for the suggestion. We will add a table with the parameter ranges to the Supplement.

Finally, they should argue why although different parameters of the two models are changed, the model outcomes can still be compared.

Regarding the chosen parameter ranges, we will add more information on that in a revised manuscript. Regarding the fact that different models have different parameters, we think that's

inevitable when working with different models. Different models will have different parameters and sometimes even if they have the same name they might actually have a different meaning.

My second major concern is about the signatures, they have a low accuracy and are not widely applicable. First, I will address the accuracy of the signatures. To determine the phase shift and amplitude ratio, a sine function is fitted on the climate forcing (PETp) and the streamflow. The method of fitting a sine function through the forcing and streamflow time series does not seem adequate to me. Most of the catchment regimes do not show a clear sinusoidal yearly cycle. This is well visible in the 16 different regime types, after Weingartner and Aschwanden (1992). For example, catchments that show two discharge peaks in one year cannot be described well by a sine function, this will lead to an error in the phase shift. In the paper two examples are given were a sinusoidal function is fitted on the climate forcing and streamflow. The timing of the sine function (phase) on the forcing and streamflow seems to be quite good in these cases. However, the sine function does not follow peak discharge and low discharges. This is clearly visible in the middle figure of the East Avon at Upavon catchment, the discharge peaks in 2001 and 2003 are not represented in the sine function (discharge is double the fitted discharge!), also the sine function does not follow the discharge in 2005/2006 when there is a low discharge. This shows that the sine fitting leads to errors in the amplitude ratio. Since the signatures are used for model evaluation, these errors could also lead to errors in the outcome of the model evaluation.

Linear regression is a commonly applied technique to extract sinusoidal components (Fourier modes) from time series (see e.g. Kirchner, 2016). The comparison between the two techniques shown in the Supplement shows the robustness of the sine wave extraction. For the method to be applicable, the time series does not have "to look like a sine curve", the sine curve is rather a description of just the average seasonal behaviour. So, the fitted sine wave is not intended to represent all the variability. The extremely high and low peaks visible in the East Avon are mostly caused by a multi-annual mode (~7 years, see also Rust et al., 2019) and hence cannot be captured by a sine wave describing the annual mode. We also refer to Referee #2 here whose suggestion might help to clarify that: "In order to analyze the seasonality relations between input [...] and output we represent the two time series by their seasonal (annual) Fourier mode." We will try to clarify that in a revised version of the manuscript.

Furthermore, the use of the potential evapotranspiration (ETp) leads to errors (and thus lower accuracy) in the signatures for semi-arid and arid catchments, since the potential evapotranspiration deviates from the actual evapotranspiration in these areas. This problem is mentioned by the authors in line 380-385. This problem can be solved by including a model to estimate the actual evapotranspiration (also mentioned by the authors). This would also help interpreting the signatures with the catchment characteristics. For example, in line 436-438 the authors state that the signatures of the US catchments show a relation with the moisture index. This conclusion is made based on visual interpretation of figure 7a. But I think this conclusion is not valid because the signatures of the dry catchments on the left side have a large uncertainty because of the use of ETp instead of ETact.

We agree that the signatures are unreliable in arid catchments and we also state that in the manuscript. We will try to emphasise this limitation more clearly in a revised version of the manuscript.

The other disadvantage of the signatures is that they are not widely applicable. The problem of using the signatures for arid and semi-arid areas is already mentioned, but this could be solved by using the actual evapotranspiration. However, these signatures are also not valid for catchments with

precipitation falling as snow. Since catchments with precipitation as snow show a typically seasonal cycle, the need of leaving these out of consideration is a major lack of the proposed signatures.

Snow, while undeniably important, is a fundamentally different process and we want to avoid conflating different processes. The seasonal cycle of a snow-dominated catchment does have a distinct seasonal pattern, but would not be well modelled by the approach we have taken here. For an alternative, see Woods (2009).

Furthermore, the signatures are also not valid for climates with a less distinct seasonal pattern, so this will further limit the applicability of the signatures. Because the signatures can only be used for a certain type of catchments, it is the question whether they contain new information on the streamflow seasonality of these catchments. There are already hydrological signatures that describe the response of streamflow to climatic forcing, for example the flow duration curve. A steep slope in the flow duration curve indicates a fast response of the streamflow to climate forcing whereas a flatter curve indicates a relatively damped response and higher storage (Yadav, 2007). Only the timing component might add new information, but since the method of determining the phase shift is not accurate, I do not see the added value.

We accept that the signature is limited to particular climates. While universal signatures applicable to every catchment seem desirable, we don't think that's realistic. In practice, using a specific signature to target specific processes that occur in specific places seems unavoidable. If the proposed signatures help us to better understand humid, non-snowy catchments (e.g. most of the UK), they still have the potential to add valuable information.

The last thing I would like to point out is that all conclusions based on visual interpretation instead of statistical analysis. For the sine fitting method, I would like to see the goodness of it or the sum of squared errors (SS), to know how well the fit of the sine function to the climatic forcing and streamflow is.

As noted above, the purpose of the sine curve is not to capture all variability in the signals, just to extract the seasonal component. Comparing the extracted sine wave with the observed time series via a goodness of fit measure will only be of limited use. As described before, the sine wave is not (and it is not intended to be) a particularly good description of the whole hydrograph. So, in catchments where the seasonal mode will explain most of the variability, we will get a "good fit" and in catchments where the seasonal mode explains little of the variability, we will get a "bad fit". But this will not tell us whether the extraction of the annual mode is robust or not. To test that, we have used two different methods and we have compared the results from two different time periods as shown in the Supplement.

For the relationships between the signatures and catchment characteristics, it would be better to calculate the correlation coefficient instead of only the visual interpretation, since this might be misleading. The same goes for the model evaluation method, it would be nice to have a quantitative statement on how well the model works. This would also make it easier to compare more models, as mentioned before.

Thank you for the suggestion. We will add tables with correlation coefficients for Figures 6 and 7. We primarily use figures as they can show us complex patterns between three variables and allow us to compare the observed signatures to the theoretical results from Figures 1-3. This would not be possible just with correlation coefficients, which have their own drawbacks (for example, the Pearson correlation as a measure of linear correlation cannot describe non-linear relationships).

Minor issues and typo's:

Minor issue 1: Line 68-70 "All of these ... streamflow seasonality." I am not convinced. For example the slope of the flow duration curve can say something about the translation of climatic forcing into streamflow seasonality. A steep slope in the flow duration curve indicates a fast response of the streamflow to precipitation inputs whereas a flatter curve indicates a relatively damped response and higher storage (Yadav, 2007).

Yes, we agree that the FDC can say something about the responsiveness of a catchment, but the FDC has its own limitations (see e.g. McMillan et al., 2017). It combines multiple hydrological processes which limits its interpretability and it doesn't yield an explicit time scale such as the phase shift.

Minor issue 2: Line 94-96 "The amplitude might ... seasonal component alone." Why stating this if it is not done for this research, is it a follow up research suggestion? Then it should be placed in the discussion.

This is just a comment that relates the signatures to other metrics existing in the literature. It is not essential, so we will remove it from the manuscript.

Minor issue 3: "catchment form" can better be replaced by catchment characteristics (For example in line 100). Catchment form suggest you are looking at the effect of a small river with a lot of branches or a stretched river.

Catchment form as defined in Wagener et al. (2007) relates to "drainage area, average basin slope, pedology, and geology". It is a commonly used term and we would prefer to stick with it.

Minor issue 4: The aim could be stated much more clearly, "test whether the seasonal signatures are useful for modelling practice (line 101)" not specific enough.

We will revise that paragraph and add more details in a revised version of our manuscript.

Minor issue 5: Line 110/111 "We use Ep ... would be needed." Not a valid argument, how much would the uncertainty increase if you add another model?

We do think that it is a valid argument. Another model would introduce uncertainty in both choosing the model and potentially choosing parameter values.

Minor issue 6: In line 124 a small remark is made on the method of the sine fitting. This could be elaborated a bit more. Why use the sine fitting method? Which methods did you compare and why did you choose for the linear regression method (it is now in the supplement, but I think it is better to include it in the text)?

We decided to report details on the sine wave fitting in the SI to make the methods section more concise. We will clarify the use of the fitting methods in a revised manuscript.

Minor issue 7: A reference is needed to support line 200 "The upper limit...shape parameter equal to 2."

We will add a reference.

Minor issue 8: About figure 3, could you explain the form of the curve when ?2 becomes larger and ?1 and fraction going to second reservoir are constant.

Let's first look at the red line in Figure 3(a) from right (black line) to left. tau_1 is always 1d, the fraction going into the second reservoir is 0.3, and tau_2 starts with a value of 10d and then

increases. So at first, both reservoirs are rather fast and we get a high amplitude ratio and a small phase shift for the outgoing sine wave (which is a mixture of the sine wave coming out of the first and the second reservoir, see Eq. 14 and 15). Then, the second reservoirs gets slower, leading to a decrease in amplitude ratio and an increase in phase shift. As the second reservoirs gets slower and slower, it will contribute less and less to the overall sine wave. For very high values of tau_2 (10000d), the sine wave coming out of the second reservoir is almost a straight line (the amplitude ratio is close to 0), so the overall sine wave is primarily consisting of the sine wave coming out of the fast reservoir. Since only 70% of the total input went into the first reservoir, we will get a sine wave that's 0.7 times the original amplitude with a very small phase shift, as the first reservoir hardly attenuates the signal. We will try to clarify that in a revised version of the manuscript.

Minor issue 9: Line 235, explain the choice for Latin Hypercube sampling.

Latin Hypercube sampling is an efficient method (Cheng and Druzdzel, 2000) that assumes uniform prior parameter distributions, which we think is adequate for the present case.

Minor issue 10: Table 1, add more information on range variables. For example for moisture index: -1= most arid and 1= most humid.

We will clarify how these indices have to be interpreted in a revised version of our manuscript.

Minor issue 11: Figure 4: add color indication to description, climatic forcing (blue) and streamflow (orange).

We will add colour indications to the figure caption.

Minor issue 12: Figure 5: Based on what criteria are the benchmark catchments chosen (grey dots)? Same goes for the two red dots, random or do they represent a certain type of catchments?

The benchmark catchments are described in Harrigan et al. (2018). The two red dots are chosen arbitrarily based on their contrasting streamflow regimes.

Minor issue 13: Line 284, missing reference to table 1. Catchment attributes

Minor issue 14: Line 300, missing reference to table 1. Catchment attributes

We will add the references.

Minor issue 15: Line 304-305 "Yet generally, ...in figure 6)." Statement is not explained in discussion, why are the US phase shift larger than for the UK catchments?

We do discuss the extremely large phase shifts in lines 457-469. These phase shifts are unreliable because these catchments are very arid. For the other catchments in the US, we couldn't find catchment attributes that could explain all the observed behaviour, which is discussed in lines 447-456. So the answer to that question is that we don't know (yet).

Minor issue 16: Figure 9b, Higher probability for high BFI for GR4J than IHACRES, but GR4J lower phase shift (max 60 days)!! Why? I would expect a larger phase shift when a larger part of the flow is slow flow.

Yes, we agree here. We would expect high BFIs to be associated with small phase shifts, but that doesn't seem to be the case here. It might have to do with the internal parametrisation of GR4J.

Minor issue 17: Line 393-395 "Since the BFI... seasonal signatures." I do not agree, the BFI cannot be used as a cause for observed patterns, but it can be related to the observed pattern. A higher base flow means more slow flow so this could be related to a larger phase shift.

Yes, the BFI cannot be seen as a cause for the observed patterns, and that's what we've written in lines 393-395.

Typo's: Line 17: sensitive Line 64: minimum Line 73: seasonality Line 278: reproduce Line 496: outputs

Thank you for pointing out these typos and thank you again for reviewing our manuscript!

References

Addor, N. and Melsen, L.A., 2019. Legacy, rather than adequacy, drives the selection of hydrological models. Water Resources Research, 55(1), pp.378-390.

Cheng, J. and Druzdzel, M.J., 2000, May. Latin hypercube sampling in Bayesian networks. In FLAIRS Conference (pp. 287-292).

Harrigan, S., Hannaford, J., Muchan, K. and Marsh, T.J., 2017. Designation and trend analysis of the updated UK Benchmark Network of river flow stations: the UKBN2 dataset. Hydrology Research, 49(2), pp.552-567.

Kirchner, J.W., 2016. Aggregation in environmental systems–Part 1: Seasonal tracer cycles quantify young water fractions, but not mean transit times, in spatially heterogeneous catchments. Hydrology and Earth System Sciences, 20(1), pp.279-297.

Knoben, W.J., Freer, J.E., Fowler, K.J., Peel, M.C. and Woods, R.A., 2019. Modular Assessment of Rainfall–Runoff Models Toolbox (MARRMoT) v1. 2: an open-source, extendable framework providing implementations of 46 conceptual hydrologic models as continuous state-space formulations. Geoscientific Model Development, 12(6), pp.2463-2480.

McMillan, H., Westerberg, I. and Branger, F., 2017. Five guidelines for selecting hydrological signatures.

Rust, W., Holman, I., Bloomfield, J., Cuthbert, M. and Corstanje, R., 2019. Understanding the potential of climate teleconnections to project future groundwater drought.

Vogel, R.M. and Sankarasubramanian, A., 2003. Validation of a watershed model without calibration. Water Resources Research, 39(10).

Wagener, T., Sivapalan, M., Troch, P. and Woods, R., 2007. Catchment classification and hydrologic similarity. Geography compass, 1(4), pp.901-931.

Woods, R.A., 2009. Analytical model of seasonal climate impacts on snow hydrology: Continuous snowpacks. Advances in water resources, 32(10), pp.1465-1481.

---

## Author Response (AR2)

Dear Martijn,

Thank you very much for handling our manuscript and for the useful suggestions. We incorporated the suggestions and

1) we now refer to the specific section in the supplement within which we changed the headers and

2) we added the suggested arrows and the values of $p$ in Figures 2 and 3.

Kind regards,

Sebastian